# Dynamics of compartment-specific proteomic landscapes of hepatotoxic and cholestatic models of liver fibrosis

**Marketa Jirouskova[1]\*, Karel Harant[2], Pavel Cejnar[3], Srikant Ojha[1,4], Katerina Korelova[1], Lenka Sarnova[1], Eva Sticova[5,6], Christoph H Mayr[7], Herbert B Schiller[7,8], Martin Gregor[1]\***

[1]Laboratory of Integrative Biology, Institute of Molecular Genetics of the Czech Academy of Sciences, Prague, Czech Republic; [2]Laboratory of Mass Spectrometry, BIOCEV, Faculty of Science, Charles University, Prague, Czech Republic; [3]Department of Mathematics, Informatics and Cybernetics, University of Chemistry and Technology, Prague, Czech Republic; [4]Department of Animal Physiology, Faculty of Science, Charles University, Prague, Czech Republic; [5]Clinical and Transplant Pathology Centre, Institute for Clinical and Experimental Medicine, Prague, Czech Republic; [6]Department of Pathology, The Third Faculty of Medicine, Charles University and University Hospital Kralovske Vinohrady, Prague, Czech Republic; [7]Helmholtz Munich, Research Unit Precision Regenerative Medicine; Comprehensive Pneumology Center, Member of the German Center for Lung Research (DZL), Munich, Germany; [8]Institute of Experimental Pneumology, LMU University Hospital, Ludwig-Maximilians University, Munich, Germany

**\*For correspondence:**
marketa.jirouskova@img.cas.cz (MJ);
martin.gregor@img.cas.cz (MG)

**Competing interest:** The authors declare that no competing interests exist.

## eLife Assessment

This **important** study suggests that the composition of the extracellular matrix in a mouse model of liver fibrosis changes depending on the cause of liver fibrosis. The data could be used as a foundation for future antifibrotic therapies. The strength of evidence is **convincing** with respect to the use of animal models and proteomic analysis. The study provides a helpful inventory of proteins up or down-regulated.

**Abstract** Accumulation of extracellular matrix (ECM) in liver fibrosis is associated with changes in protein abundance and composition depending upon etiology of the underlying liver disease. Current efforts to unravel etiology-specific mechanisms and pharmacological targets rely on several models of experimental fibrosis. Here, we characterize and compare dynamics of hepatic proteome remodeling during fibrosis development and spontaneous healing in experimental mouse models of hepatotoxic (carbon tetrachloride [$CCl_4$] intoxication) and cholestatic (3,5-diethoxycarbonyl-1,4-dihydrocollidine [DDC] feeding) injury. Using detergent-based tissue extraction and mass spectrometry, we identified compartment-specific changes in the liver proteome with detailed attention to ECM composition and changes in protein solubility. Our analysis revealed distinct time-resolved $CCl_4$ and DDC signatures, with identified signaling pathways suggesting limited healing and a potential for carcinogenesis associated with cholestasis. Correlation of protein abundance profiles with fibrous deposits revealed extracellular chaperone clusterin with implicated role in fibrosis resolution. Dynamics of clusterin expression was validated in the context of human liver fibrosis. Atomic force microscopy of fibrotic livers complemented proteomics with profiles of disease-associated changes in local liver tissue mechanics. This study

determined compartment-specific proteomic landscapes of liver fibrosis and delineated etiology-specific ECM components, providing thus a foundation for future antifibrotic therapies.

## Introduction

Liver fibrosis and subsequent cirrhosis, two leading causes of liver disease-related deaths worldwide (*Karlsen et al., 2022*), develop as a result of chronic liver injury of multiple etiologies (*Kisseleva and Brenner, 2021*). Fibrosis is characterized by excessive accumulation of extracellular matrix (ECM) proteins forming fibrous scar tissue. In the liver, such pathological matrix is deposited by activated hepatic stellate cells (HSCs) and/or portal fibroblasts (PFs) in response to the inflammatory reaction (*Kisseleva and Brenner, 2021*; *Henderson et al., 2020*). Recent multi-omics studies have framed fibrosis as a dynamic multicellular process associated with the remodeling of gene expression landscapes (*Ramachandran et al., 2019*) and profound changes in liver protein abundance as well as composition (*Niu et al., 2022*; *Massey et al., 2017*; *Klaas et al., 2016*).

Hepatocellular injury alters parenchymal mechanical properties (*Georges et al., 2007*), thus further facilitating activation of resident HSCs and PFs and their differentiation into myofibroblasts depositing collagen-rich ECM (*Olsen et al., 2011*). Accumulation of abnormal ECM further promotes ECM-depositing myofibroblasts and fosters progressive whole-organ stiffening as a result of ongoing fibrogenesis (*Saneyasu et al., 2016*). Thus, ECM-associated changes are increasingly perceived as causative, rather than consequential, and multiple efforts have focused on the identification of extracellular niche components that drive the pathogenesis of fibrosis (*Schiller et al., 2015*).

The ECM is a complex network of hundreds of proteins acting as a three-dimensional scaffold, supporting cells mechanically and functioning as a reservoir for secreted factors (e.g. growth factors and cytokines) (*Hynes, 2009*). This so-called 'matrisome' consists of 'core matrisome' (the structural components of the ECM, i.e. collagens, glycoproteins, and proteoglycans) and 'matrix-associated matrisome' (the ECM-interacting proteins) (*Naba et al., 2012*). Transitions of extracellular factors between the insoluble matrisome and the soluble pools modulate their signaling capabilities and bioavailability (*Schiller et al., 2015*). As fibrous ECM assemblies also determine tissue mechanics, matrisome constituents provide matrix-embedded cells with spatially distinct biochemical and biomechanical context.

In terms of etiology, chronic liver injury is inflicted either by hepatotoxic or by cholestatic insult (*Kisseleva and Brenner, 2021*). In recent decades, several corresponding animal models of liver fibrosis have been established (*Liu et al., 2013*). Despite several translational limitations (*Trautwein et al., 2015*), these experimental models mimic fundamental aspects of human liver fibrosis (*Liu et al., 2013*). Thus, frequently used iterative carbon tetrachloride (CCl$_4$) intoxication causes hepatocellular damage, HSC activation, and development of pericentral liver fibrosis that evolves into severe bridging fibrosis (*Liu et al., 2013*). In contrast, 3,5-diethoxycarbonyl-1,4-dihydrocollidine (DDC) feeding induces obstructive cholestasis, characterized by expansion of PFs and consecutive periportal fibrosis with typical portal-portal septa (*Fickert et al., 2007*). As both human and animal studies have shown that liver fibrosis can be ameliorated by either targeting progression or promoting resolution (*Trautwein et al., 2015*), experimental models have become instrumental for identifying factors and mechanisms central to blocking fibrosis progression and promoting the reversal of advanced fibrosis.

In this comparative study, we employed CCl$_4$- and DDC-based mouse models to describe proteomic changes during liver injury, fibrosis development, and repair while focusing on matrisome remodeling and associated alterations in tissue biomechanics. Our in-depth analysis of mass spectrometry (MS) data obtained from total liver lysates and ECM-enriched insoluble fractions defines compartment- and time-resolved proteomic signatures in hepatotoxin- vs. cholestasis-induced fibrosis and healing and delineates disease-specific matrisome.

**eLife digest** Alcoholism or chronic conditions like hepatitis damage the liver. Over time, scar tissue builds up in the liver, causing cirrhosis. The scaring results from the liver's repeated attempts to repair itself by creating more structural proteins called extracellular matrix proteins. A buildup of these scaffolding proteins leads to tissue stiffening or fibrosis. Fibrosis may heal in some cases but in others, it may progress to cirrhosis, liver cancer or liver failure.

Learning more about these processes may help scientists and clinicians understand why fibrosis is reversible in some cases but not others. It may also allow them to develop treatments that can treat or reverse fibrosis and prevent cirrhosis, liver cancer, or liver failure. The first step is studying how fibrosis occurs in mouse models that mimic different types of liver disease. For example, repetitive ingestion of a toxic substance, such as alcohol, can cause one type of liver disease. However, slowing or stalling bile flow through the biliary system (the liver, gallbladder, and bile ducts), leads to a different type of chronic liver injury.

Jirouskova et al. identify an extracellular protein called clusterin that may help heal fibrosis. The experiments used mouse models of two different types of liver disease. One mimicked liver disease caused by repetitive toxin injury, and the other modelled liver disease caused by chronic stalling of the bile flow in the liver (cholestasis). In the experiments, Jirouskova et al. looked at all the proteins made in each type of liver disease as the animals developed fibrosis or their fibrosis resolved. They also studied extracellular matrix proteins and how they affected molecular signaling in the liver tissue. The experiments revealed different patterns of protein production and healing in the different types of liver disease. The animals with liver diseases caused by chronic cholestatic injury were less likely to heal their livers and showed potential to progress to liver cancer. Production of the clusterin protein was connected with better liver recovery from toxic injuries.

Jirouskova et al. provide a comprehensive map of all the proteins produced over the course of liver fibrosis progression and healing in two different animal models of liver disease. Scientists and clinicians may use this information to study liver disease types. It may also one day help them personalize patient's therapies. The experiments show that extracellular matrix proteins are essential contributors to fibrosis and key signaling agents in liver disease. This may make them good targets for new therapies. Boosting clusterin production may be one approach to promoting liver recovery. More studies are needed to confirm this before such therapies can be developed and tested in humans.

## Results

### Proteomic analysis outlines the gradual fibrosis development and partial healing in both CCl$_4$- and DDC-induced liver injury

To compare time-resolved, compartment-specific proteomes of CCl$_4$- and DDC-induced models of liver fibrosis, total liver lysates (Total) together with two protein fractions obtained by one-step detergent extraction (ECM-enriched insoluble fraction [E-fraction] and soluble fraction [S-fraction]) were prepared for MS analysis (*Figure 1A*). The gradual fibrogenesis was assessed at two time points (T1 and T2) corresponding to mild fibrosis with partially bridging septa (T1, 3 weeks for CCl$_4$ and 2 weeks for DDC treatment) and advanced fibrosis (T2, 6 weeks for CCl$_4$ and 4 weeks for DDC). To characterize the dynamics of spontaneous fibrosis resolution, we allowed the mice 5 (T3) or 10 days (T4) to recover. Both models developed typical progressive liver damage with fibrotic scarring followed by partial healing upon insult withdrawal as demonstrated by plasma levels of liver injury markers and the extent of collagen-rich deposits in liver sections (*Figure 1—figure supplement 1*).

Using the two models, we quantified 4734 proteins approximately evenly distributed across all time points and fractions (*Figure 1B*). The numbers of quantified proteins were comparable across time and treatments with proteins equally represented in each of the cohorts. No bias was shown against any of the analyzed cohorts or time points (*Figure 1B*). Likewise, the numbers of quantified proteins across fractions showed only minor variability (*Figure 1B*). We identified 184 matrisome constituents (*Naba et al., 2012*; *Shao et al., 2020*) among the quantified proteins. Their median sequence coverage in controls, CCl$_4$, and DDC samples only slightly differed from sequence coverage of all identified proteins, showing no strong bias against matrisome-annotated proteins (*Figure 1C*).

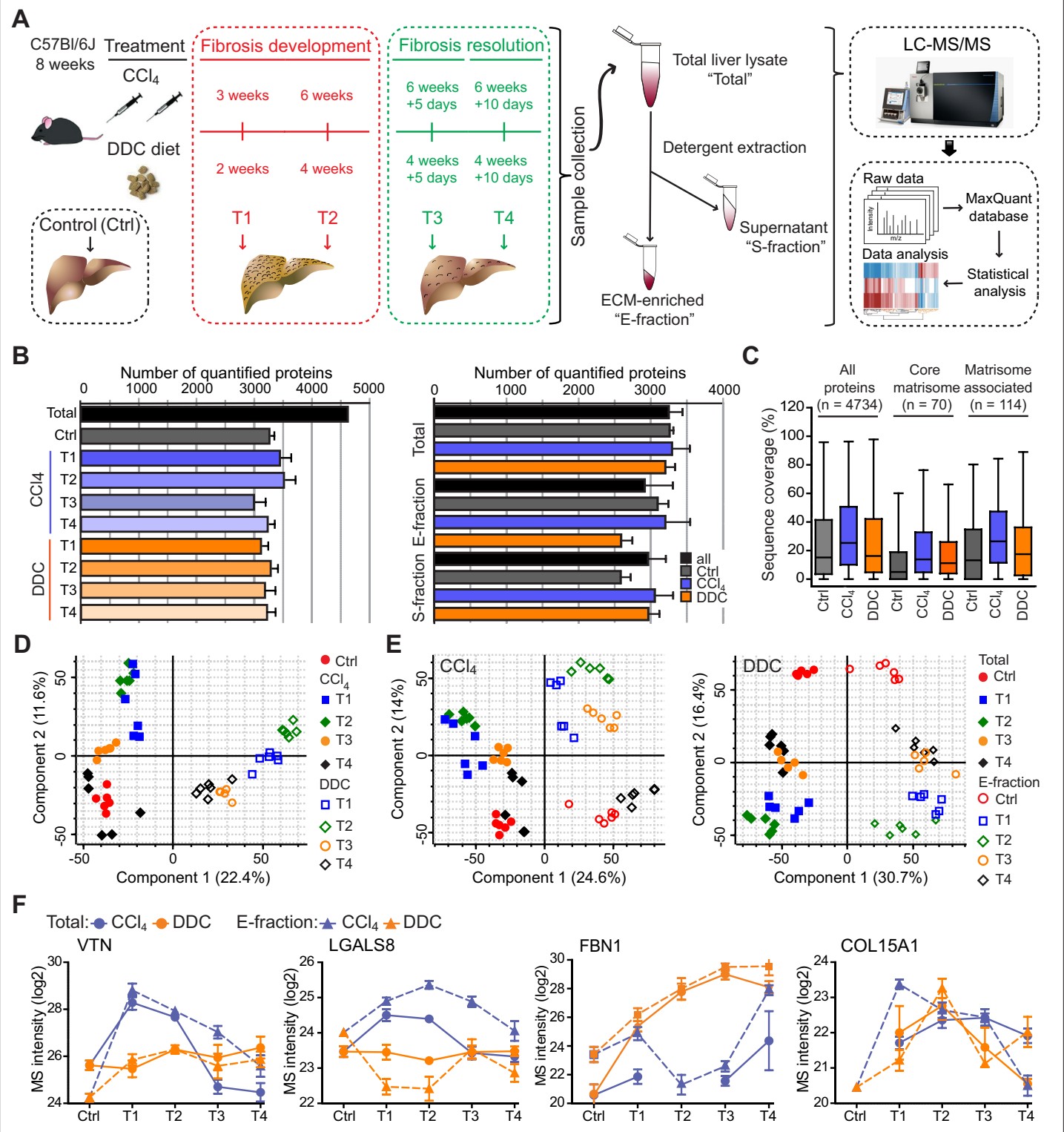

**Figure 1.** Hepatotoxic and cholestatic injury generate distinct time-resolved and compartment-specific protein signatures. (**A**) Schematic overview of the experimental setup. Six animals were used in each model at each time point. (**B**) Numbers of quantified proteins at the indicated time points and experimental conditions; n=6. (**C**) Box plot shows the distribution of protein sequence coverage (coverage of possible tryptic peptides per protein in %) for the indicated matrisome categories (as defined by *Naba et al., 2012*) and all detected proteins in experimental conditions indicated. (**D**) Principal component analysis (PCA) of Total proteome separates time-dependent fibrogenesis and healing in carbon tetrachloride (CCl₄)- (closed symbols) and 3,5-diethoxycarbonyl-1,4-dihydrocollidine (DDC)- (open symbols) induced fibrosis. The first two components of data variability of 3624 proteins

*Figure 1 continued on next page*

*Figure 1 continued*

identified in Total liver fractions in CCl$_4$ and 3521 proteins in DDC are shown; n=6. (**E**) PCA shows the separation of Total (closed symbols) and E-fraction (open symbols) proteomes in time; the first two components of data variability are shown; n=6. (**F**) Line plots show time-dependent changes in mass spectrometry (MS) intensities in Total (solid line) and E-fraction (broken line) proteomes for indicated selected proteins in CCl$_4$ and DDC models; n=4–6.

The online version of this article includes the following figure supplement(s) for figure 1:

**Figure supplement 1.** Liver damage markers and collagen deposits in carbon tetrachloride (CCl$_4$) and 3,5-diethoxycarbonyl-1,4-dihydrocollidine (DDC) models.

**Figure supplement 2.** Hierarchical cluster analysis shows the separation of carbon tetrachloride (CCl$_4$)- and 3,5-diethoxycarbonyl-1,4-dihydrocollidine (DDC)-derived Total proteomes.

Principal component analysis (PCA) revealed clear temporal separation of CCl$_4$- and DDC-derived samples in both Total and E-fraction (*Figure 1D and E*). Samples also separated in time, which demonstrated fibrosis development and its partial resolution for Total and E-fraction samples in both models (*Figure 1E*).

Unsupervised hierarchical cluster analysis of total lysate proteins together with annotation enrichment of the observed clusters demonstrated substantial differences in the dynamic regulation of protein expression between the two models (*Figure 1—figure supplement 2*). This was evidenced by differential Total MS protein intensity temporal profiles of matrisome-enriched cluster constituents, such as vitronectin (VTN), galectin 8 (LGALS8), fibrillin 1 (FBN1), or a putative portal myofibroblast marker collagen type XV α-1 chain (COL15A1; *Figure 1F*). Moreover, distinct galectin 8 and vitronectin expression profiles obtained for CCl$_4$ and DDC E-fractions revealed complex changes in protein association with the insoluble proteome. Taken together, our MS data allow for time-resolved identification of proteins differentially expressed during fibrosis progression and resolution in etiologically distinct models of liver fibrosis. Moreover, this approach enables analysis of complex changes in the solubility of the identified proteins and their transition between liver compartments.

## Time-resolved analysis of Total proteomes indicates limited healing and potential tumorigenicity in the DDC model

To capture the proteome differences between hepatotoxin- vs. cholestasis-induced fibrogenesis, we determined proteins significantly regulated during fibrogenesis separately in CCl$_4$ and DDC Total samples (Benjamini-Hochberg false discovery rate [BH FDR]<5%; see Materials and methods). In total, 762 proteins were found to be shared by the two models, while 514 (CCl$_4$) and 1074 (DDC) proteins were identified as unique for the respective model (*Figure 2A*).

Using Ingenuity Pathway Analysis (IPA), we predicted upstream transcriptional regulators and growth factors together with corresponding downstream biological signaling pathways differentially associated with identified protein groups (*Figure 2B*, *Figure 2—figure supplement 1A*). Abundance changes in known targets indicated the activity of key regulators involved in fibrogenesis (transforming growth factor beta-1 [TGFB1] and angiotensinogen [AGT]), hypoxia response (hypoxia-inducible factor 1-alpha [HIF1A]), and inhibition of factors mediating hepatocyte function (hepatocyte nuclear factor 4-alpha [HNF4A] and 1-alpha [HNF1A]) or involved in HSC inactivation (peroxisome proliferator-activated receptor gamma coactivator 1-alpha [PPARGC1A]) in both fibrotic models (*Figure 2—figure supplement 1A*). Interestingly, the pro-proliferative mammalian target of rapamycin (mTOR) pathway showed signaling attenuation in the course of CCl$_4$ treatment but upregulation in the case of DDC treatment, suggesting hyperplastic potential of the DDC model (*Figure 2—figure supplement 1A*). Consistently, analysis of the DDC signature revealed inhibition of tumor suppressor gene *Rb1* with simultaneous activation of potential oncogenic regulators lysine demethylase 5A (KDM5A), myocardin-related transcription factor A (MRTFA), myelocytomatosis oncogene (MYC), and yes-associated protein (YAP), thus underscoring a potential carcinogenic risk of cholestatic model (*Figure 2B*). By contrast, the CCl$_4$ signature indicated activation of epidermal growth factor (EGF) and vascular endothelial growth factor A (VEGFA) with simultaneous downregulation of the HNF4A (*Figure 2B*), thus supporting previous reports on the importance of these factors during hepatotoxic injury (*Kömüves et al., 2000*; *Yang et al., 2014*; *Yue et al., 2010*). While dynamic changes of regulatory networks and signaling pathways correlated well with fibrosis progression in both models,

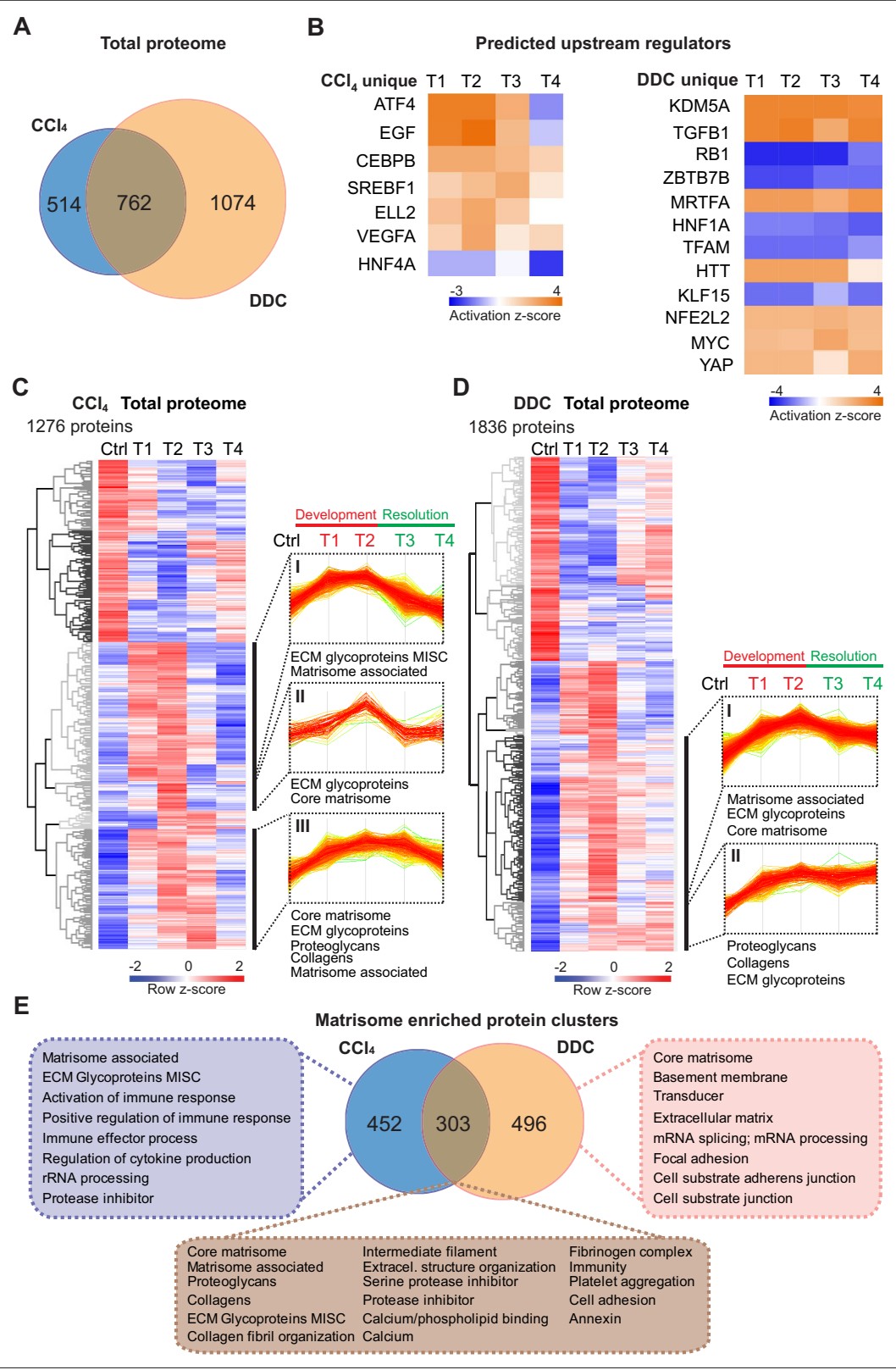

**Figure 2.** Time-resolved analysis of Total proteomes shows limited healing in matrisome-enriched protein clusters in the cholestatic model. (**A**) Venn diagram shows relative proportion of 1276 and 1836 proteins differentially expressed (t-test; Benjamini-Hochberg false discovery rate [BH FDR]<5%) in Total carbon tetrachloride (CCl₄) and 3,5-diethoxycarbonyl-1,4-dihydrocollidine (DDC) proteomes, respectively. (**B**) Hierarchical cluster analysis of the

*Figure 2 continued on next page*

*Figure 2 continued*

activity score of the upstream regulators at the indicated time points predicted by Ingenuity Pathway Analysis (IPA) from unique CCl$_4$ and DDC protein signatures shown in A. (**C, D**) Hierarchical clustering of mean $z$-scored mass spectrometry (MS) intensities of proteins of Total CCl$_4$ (C) or DDC (D) proteomes; n=6. Profiles of $z$-scored MS intensities of proteins from matrisome-annotated clusters for CCl$_4$ (I–III) and DDC (I and II) models are shown. (**E**) Venn diagram compares proteins from matrisome-annotated clusters shown in C and D. UniProt keyword enrichment annotation for each group within the diagram is indicated (Fisher's test, BH FDR<4%).

The online version of this article includes the following figure supplement(s) for figure 2:

**Figure supplement 1.** Ingenuity Pathway Analysis (IPA) of upstream regulators and canonical signaling pathways in carbon tetrachloride (CCl$_4$) and 3,5-diethoxycarbonyl-1,4-dihydrocollidine (DDC) models.

**Figure supplement 2.** Collagens and laminins identified in Total proteomes.

ineffective downregulation of profibrotic signaling in the DDC-induced model suggested a limited capacity of the liver to heal from cholestatic injury (*Figure 2B*, *Figure 2—figure supplement 1*).

Next, we analyzed CCl$_4$ and DDC Total proteomes using unsupervised hierarchical clustering and functional annotation term enrichment (*Figure 2C and D*). This allowed the separation of proteins with similar temporal abundance profiles while simultaneously revealing clusters comprising matrisome-annotated proteins (see Materials and methods). In the CCl$_4$ proteome, three matrisome-annotated clusters (*Figure 2C*; clusters I, II, and III) containing 755 proteins showed a variable degree of time-dependent abundance decline over the course of healing. By contrast, two large matrisome-annotated clusters (*Figure 2D*; clusters I and II) with 799 constituents showed almost no decline in abundance over the recovery period in the DDC model. Our data reflect the reversibility of CCl$_4$-induced fibrogenesis and restricted capacity to heal with higher potential of carcinogenic risk in the DDC model.

## Matrisome linked with cholestasis is enriched in basement membrane proteins whereas deposits upon hepatotoxic injury contain matrisome-associated proteins

To further dissect disease-specific processes associated with matrisome-enriched clusters (*Figure 2C and D*), we followed the functional annotation of proteins using a comprehensive annotation matrix compiled from Gene Ontology terms, UniProtKB keywords, and the MatrisomeDB matrisome database (*Naba et al., 2012*; *Shao et al., 2020*). This revealed that 303 proteins shared by CCl$_4$ and DDC Total proteomes were annotated not only with matrisome-related keywords but also as 'cell adhesion', 'fibrinogen complex', 'platelet aggregation', and 'intermediate filaments', thus indicating engagement in cellular interaction with ECM (*Figure 2E*). Consistently, top canonical IPA pathways included 'signaling by Rho family GTPases', 'integrin signaling', and 'actin cytoskeleton signaling' (*Figure 2—figure supplement 1C*). As anticipated, 452 proteins uniquely identified in CCl$_4$ matrisome-enriched clusters were also associated with inflammatory response, a well-described feature of this model (*Ghallab et al., 2019*). A group of 496 proteins exclusive for DDC was found to associate with 'basement membrane' (BM) and 'cell substrate junction' reflecting an increased abundance of BM components synthesized during the development of periductal fibrosis (e.g. laminins, α-chains of collagen types IV, VI, and XVIII; *Figure 2E*).

Within a group of 60 matrisome proteins significantly changing in both Total CCl$_4$ and DDC proteomes (*Figure 3A*), there were mainly core matrisome proteins, such as collagens (e.g. α-chains of collagen types I and V) and ECM glycoproteins (e.g. fibronectin, EMILIN1, and vitronectin). In contrast to previous reports (*Klaas et al., 2016*), we found collagen type VI, a protein regulating ECM contractility (*Williams et al., 2020*), upregulated during fibrogenesis in both models (*Figure 2—figure supplement 2*). In addition, we identified several matrisome-associated proteins (e.g. NGLY1, PLOD3, S100A4, MUG2, SERPINA7, SERPINF1, and P4HA1) not previously reported in liver fibrosis as uniquely induced in the CCl$_4$ model (*Shao et al., 2020*). Among these, ECM glycoprotein lactadherin (MFGE8) has been shown to reduce liver fibrosis in mice (*An et al., 2017*). In contrast, BM matrisome proteins, such as laminins, collagen types IV and XVIII, and perlecan (HSPG2), were exclusively upregulated in the DDC proteome (*Figure 3A*, *Figure 2—figure supplement 2*).

To validate our findings demonstrating distinct features of the fibrotic models used, we performed immunofluorescence (IF) analysis of major core matrisome proteins abundantly upregulated in

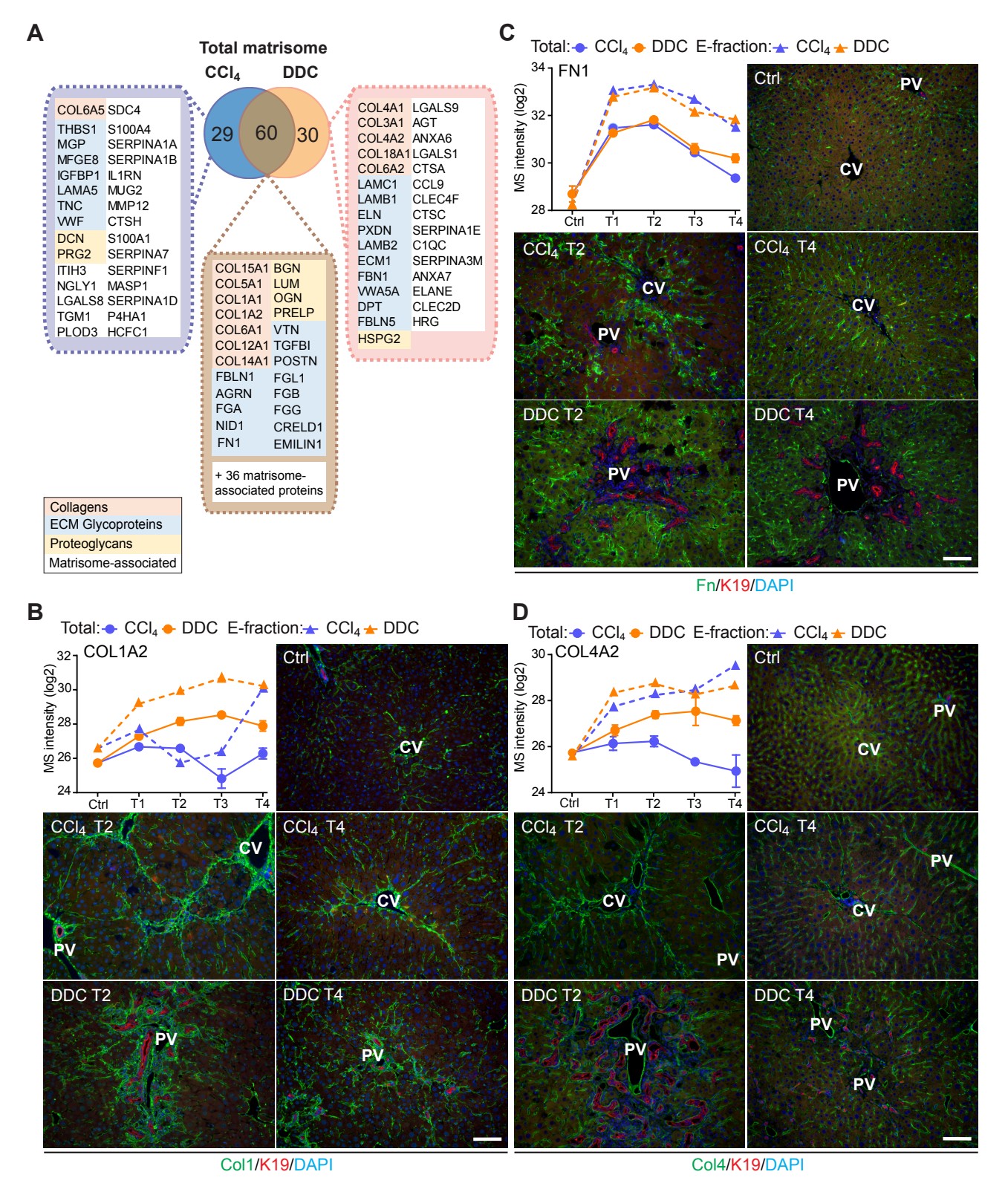

**Figure 3.** Comparison of matrisome proteins differentially expressed between the carbon tetrachloride ($CCl_4$)- and 3,5-diethoxycarbonyl-1,4-dihydrocollidine (DDC)-derived Total proteomes with immunofluorescence (IF) localization of the main core matrisome proteins within the injured livers. (**A**) Venn diagram shows a comparison of matrisome proteins differentially enriched in Total $CCl_4$ and DDC proteomes. Color coding indicates identified matrisome categories. (**B–D**) Representative IF images of collagen type I (B), fibronectin (C), and collagen type IV (D), all in green in liver sections from

*Figure 3 continued on next page*

*Figure 3 continued*

untreated controls (Ctrl), CCl₄-, and DDC-treated mice at time points of fibrosis development (T2) and resolution (T4). Bile ducts were visualized with antibodies to keratin 19 (K19; red); nuclei were stained with DAPI (blue). CV, central vein; PV, portal vein. Scale bar = 100 µm. Line plots show time-dependent change in respective mass spectrometry (MS) intensities in Total (solid line) and E-fraction (broken line) proteomes in CCl₄ and DDC models; n=4–6.

both CCl₄ and DDC proteomes (*Figure 3B–D*). Collagen type I accumulated at the sites of primary injury and, consistent with proteomics, was only partially reduced upon recovery in the DDC model (*Figure 3B*). Collagen type I-rich areas were delineated by fibronectin, a protein serving as a scaffold for collagen fibril assembly (*Sottile et al., 2007*). Although fibronectin abundance in Total CCl₄ and DDC proteomes decreased with healing, IF analysis revealed persisting deposits within capillarized hepatic sinusoids even after a recovery period of 10 days (*Figure 3C*). Consistent with a report on human cirrhotic livers (*Mak et al., 2013*), BM-associated collagen type IV accumulated around fibrotic septa in both models and its persistent presence indicated incomplete liver healing after DDC withdrawal (*Figure 3D*).

To analyze proteins that are induced during fibrosis development in Total proteome, we identified proteins detected in T2 and T1 but not in control samples in each model. This way we identified 65 newly induced proteins in CCl₄- and 118 in DDC-treated livers. These were mainly matrisome proteins and proteins of the immune defense response (*Figure 2—figure supplement 1D*). Notably, we found lysyl hydroxylase (Plod3), a matrisome protein highly expressed in activated HSCs (*Nishio et al., 2019*), among newly expressed CCl₄-specific proteins. In contrast, DDC-specific matrisomal proteins included elastin (ELN), a proposed PF marker (*Yang et al., 2021*; *Figure 2—figure supplement 1D*). This confirmed the specificity of identified CCl₄ and DDC signatures and suggested the potential of our approach in describing the different origin of collagen-producing myofibroblasts in analyzed models.

## Integrin αv is specifically induced on the membranes of injured hepatocytes within fibrotic scar tissue in the hepatotoxic CCl₄ model

The dynamic nature of fibrosis stems from interplay between injured hepatocytes, immune cells, and hepatic myofibroblasts (*Henderson et al., 2020*). Using previously published cell-type-specific signatures (see Materials and methods), we identified 10 different cell populations in CCl₄ and DDC samples and compared their abundances over the course of liver fibrosis (*Figure 4A–D*, *Figure 4—figure supplement 1A*). A set of 18 proteins quantified from the signature of hepatocytes exhibited faster decline in the hepatotoxic CCl₄ than DDC model (*Figure 4A*), corresponding well to the extensive parenchymal injury evidenced by high alanine and aspartate transaminase levels (*Figure 1—figure supplement 1A*). In support, large areas of injured pericentral hepatocytes were found negative for HNF4α staining in CCl₄ model (*Figure 4—figure supplement 2A*). Rapid increase in HSC and activated PF signatures paralleled the activation and proliferation of ECM-depositing myofibroblasts in both fibrotic models (*Figure 4B*) evidenced by IF staining of myofibroblast marker alpha smooth muscle actin (αSMA; *Figure 4—figure supplement 2A*). However, faster upregulation of activated PFs in DDC confirms their predominant role in cholestatic fibrosis. Temporal abundance profiles of Kupffer cell, macrophage, monocyte, and B-cell signature proteins illustrate faster recovery from inflammation with healing in the CCl₄ than in DDC model (*Figure 4C and D*, *Figure 4—figure supplement 1A*). This was further confirmed by IF staining of liver sections (*Figure 4—figure supplement 2B*). These findings, together with matrisome-enriched protein cluster analysis (*Figure 2E*), emphasized the role of the inflammation component in the context of fibrosis and underscored the model-specific involvement of PFs in cholestatic fibrosis.

Cellular interactions with the changing microenvironment are mediated by integrins, a heterogeneous family of cell adhesion receptors. Thus, temporal changes in cell-type-specific integrin subsets reflect the alterations in fibrotic injury-induced cell populations with the potential to predict the fibrogenicity of the ongoing disease. Our profiling revealed increased expression of integrin β1 (*Figure 4—figure supplement 1B*), the most abundant collagen receptor, which has been reported (together with integrins α1 and α5) to correlate with the stages of human liver fibrosis (*Nejjari et al., 2001*). Given its cell-type-specific expression prevalence, this finding likely corresponds to the expansion of HSCs, simultaneously with induced integrin β1 expression on liver sinusoidal endothelial cells

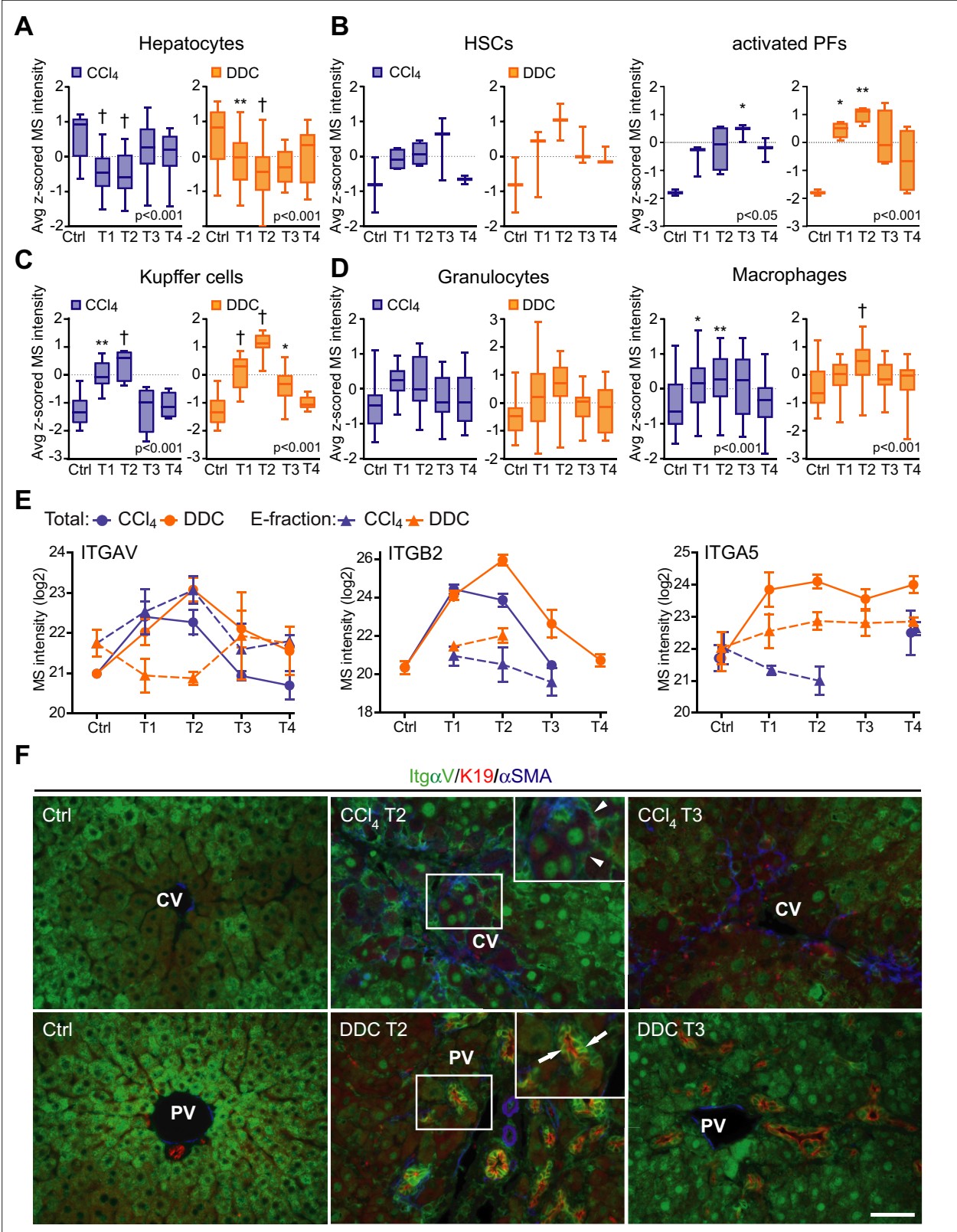

**Figure 4.** Liver cell-type dynamics and cell-type-specific integrin expression during fibrogenesis and healing. (**A–D**) Box plots show mean z-scored mass spectrometry (MS) intensities of the indicated cell-type-specific protein signatures in time. Hepatocytes (A; n=18), hepatic stellate cells (HSCs), and activated portal fibroblasts (PFs) (B; n=3 and 4), Kupffer cells (C; n=6), granulocytes, and macrophages (D; n=16 and 33). One-way ANOVA with Bonferroni's post-test; *p<0.05; **p<0.01; †p<0.001. (**E**) The line plots show the time-dependent change in MS intensities of indicated selected integrins

*Figure 4 continued on next page*

*Figure 4 continued*

in Total (solid line) and E-fraction (broken line) proteomes in carbon tetrachloride (CCl₄) and 3,5-diethoxycarbonyl-1,4-dihydrocollidine (DDC) models; n=4–6. (**F**) Representative immunofluorescence images of liver sections from untreated controls (Ctrl), CCl₄-, and DDC-treated mice at time points of fibrosis development (T2) and resolution (T3) immunolabeled for integrin αv (green), K19 (red), and αSMA (blue). Arrowheads, integrin αv-positive injured hepatocytes; arrows, integrin αv-positive biliary epithelial cells of reactive ductuli. CV, central vein; PV, portal vein. Boxed areas, ×2 images. Scale bar = 50 μm.

The online version of this article includes the following figure supplement(s) for figure 4:

**Figure supplement 1.** Dynamics of liver sinusoidal endothlial cells (LSECs) and immune cells with their cell-type-specific integrin expression profiles.

**Figure supplement 2.** Liver cell-type-specific dynamics development during fibrogenesis and healing documented by immunofluorescence microscopy corroborates the mass spectrometry (MS)-derived cell-type-specific signature dynamics.

(LSECs) and injured hepatocytes (*Nejjari et al., 2001*). A rapid immune response is documented by extensive upregulation of main leukocyte integrin subunits β2 and αM (*Figure 4*, *Figure 4—figure supplement 1*). Their differential expression kinetics between the models correlate well with cell-type-specific signatures (*Figure 4*, *Figure 4—figure supplement 1*). Unexpectedly, fibronectin receptor integrin α5 (typically expressed on HSCs, PFs, and LSECs [*Olsen et al., 2012*; *Couvelard et al., 1993*]) was detected in Total DDC proteome only (*Figure 4E*). Since a boosted expression of integrin α5 has been reported in liver tumors (*Lai et al., 2011*), this finding further supports the tumorigenic potential of biliary fibrosis.

Strong induction of TGFβ-activating integrin αv in both CCl₄ and DDC Total proteomes (*Figure 4E*) reflected its central role in fibrogenesis (*Reed et al., 2015*). Strikingly, in E-fraction, integrin αv was found to increase only in the CCl₄ model (*Figure 4E*), suggesting its close association with ECM specifically during hepatotoxic injury. Subsequent IF analysis showed localization of antibodies to αv mostly to periportal hepatocytes and with little signal found in central hepatocytes of control livers (*Figure 4F*). During CCl₄-induced injury, αv staining was enhanced at the periphery of pericentral hepatocytes adjacent to the collagen-rich scars, reflecting ongoing liver periportalization with the injury (*Ghallab et al., 2019*). The zonal distribution of αv was partially restored with fibrosis regression. In DDC-driven cholestasis, antibodies to αv stained strongly with reactive biliary epithelial cells (BECs), the main drivers of fibrogenesis in biliary fibrosis (*Figure 4F*). Thus, our analysis reveals stage-specific induction of integrin αv on the surface of pericentral hepatocytes that has been unrecognized to date and suggests its potential as a marker of reactive BECs in cholestasis.

## Analysis of solubility profile dynamics of liver proteome reveals extracellular matrix protein-1 among matrisome proteins induced by fibrogenesis

As ECM remodeling during fibrogenesis also entails changes in the solubility of its constituents and associated proteins (*Schiller et al., 2015*), we next set out to identify proteins that become increasingly insoluble with fibrosis progression. We found 1273 (CCl₄) and 762 (DDC) differentially expressed proteins, defined by at least threefold higher expression in E- than in S-fraction (see Materials and methods). Hierarchical clustering of their MS intensity ratios clearly reflected changes in protein solubility profiles in the course of disease progression (*Figure 5—figure supplement 1A and B*). Proteins grouped into three (CCl₄, 406 proteins) and two (DDC, 63 proteins) ECM-enriched clusters exhibited increasing insolubility during fibrogenesis. Matrisome proteins found in both fibrotic models were mainly collagens and ECM glycoproteins (*Figure 5A*). Interestingly, we identified 93 (CCl₄) and 16 (DDC) proteins not present in control samples but induced by treatment (*Figure 5—figure supplement 1C*). Abundance of most of the identified proteins in E-fraction decreased with healing in the CCl₄ but not in the DDC model (*Figure 5B*), suggesting model-specific incorporation of matrisome-associated proteins into insoluble ECM.

Among proteins upregulated with disease onset in E-fractions of both fibrotic models was extracellular matrix protein-1 (ECM1; *Figure 5C*, *Figure 5—figure supplement 1C*). Subsequent IF analysis confirmed weak ECM1 staining (corresponding to low expression levels) in control livers, which was confined mostly to Kupffer cells and the apical membrane of BECs (*Figure 5C*, *Figure 5—figure supplement 1D*). Increased ECM1 abundance during fibrogenesis was paralleled by its increasing localization within the infiltrating inflammatory cells and activated Kupffer cells. In addition, ECM1

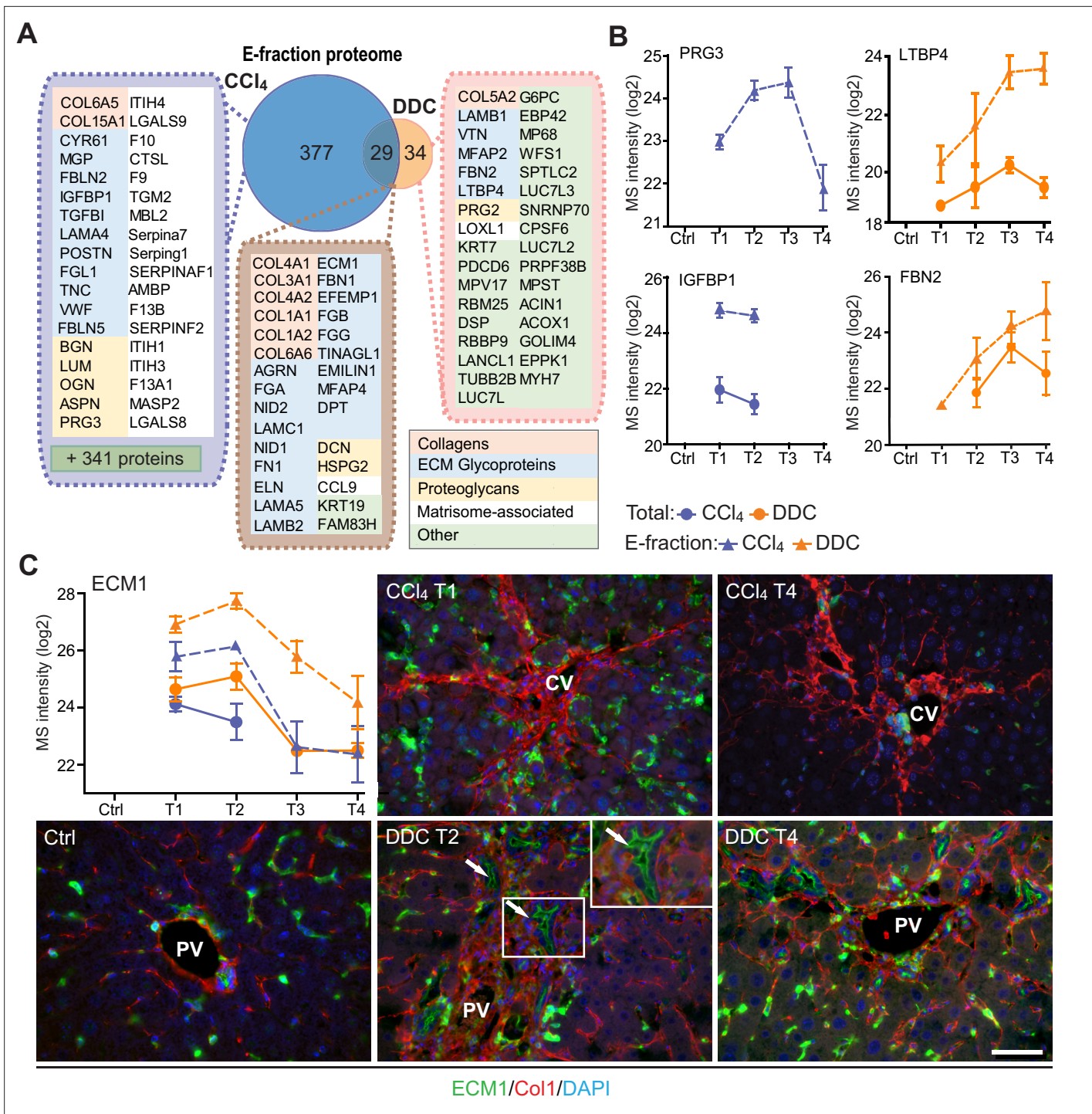

**Figure 5.** Solubility profiling provides in-depth analysis of model-specific matrisome composition. (**A**) Venn diagram shows relative proportion of proteins from E-fraction proteome identified as proteins with increasing insolubility over the course of fibrosis in each model (see *Figure 5—figure supplement 1A and B* and Materials and methods). Matrisome proteins are highlighted with color coding to indicate identified matrisome categories. (**B, C**) Line plots show time-dependent change in mass spectrometry (MS) intensities of indicated selected matrisome proteins uniquely identified in E-fraction (broken line) proteomes in carbon tetrachloride (CCl₄) and 3,5-diethoxycarbonyl-1,4-dihydrocollidine (DDC) models; n=4–6. (Solid line shows MS intensity profile in Total proteome.) (**C**) Representative immunofluorescence images of liver sections from untreated controls (Ctrl), CCl₄-, and DDC-treated mice at indicated time points of fibrosis development (T1 and T2) and resolution (T4) immunolabeled for ECM1 (green) and collagen type I (red). Nuclei were stained with DAPI (blue). Arrows, ECM1-positive reactive biliary epithelial cells. CV, central vein; PV, portal vein. Boxed areas, ×2 images. Scale bar = 50 μm.

*Figure 5 continued on next page*

*Figure 5 continued*

The online version of this article includes the following figure supplement(s) for figure 5:

**Figure supplement 1.** Analysis of matrisome proteins identified in carbon tetrachloride (CCl₄)- and 3,5-diethoxycarbonyl-1,4-dihydrocollidine (DDC)-derived E-fraction proteomes.

heavily decorated apical membranes of BECs forming reactive ductuli in DDC-treated livers, thus corresponding to higher MS intensity of ECM1 during the development of biliary fibrosis. Quantification of IF staining also showed dynamic changes in ECM1 antigen abundance during fibrosis development and resolution thus confirming the MS data (*Figure 5—figure supplement 1E*). Induction of ECM1 expression in the inflammatory cells at the sites of primary injury in the CCl₄ model and in reactive ductuli in the DDC model strongly indicates its involvement in fibrogenesis and healing. Although a previous study had identified ECM1 expression to be hepatocyte-specific (*Fan et al., 2019*), our data suggest that its role might be more complex than anticipated.

## Correlation of dynamic changes of hepatotoxic proteome and fibrotic deposits identifies clusterin as a novel protein associated with fibrosis resolution

The CCl₄ model gradually developed typical bridging liver fibrosis (*Liu et al., 2013*) followed by fibrosis resolution as documented by liver injury markers (*Figure 1—figure supplement 1A*) and collagen deposits (*Figure 1—figure supplement 1B*), as well as by results of our MS profiling (*Figure 2*). These dynamic changes allowed us to correlate protein abundance profiles (from both Total and E-fraction proteomes) with disease dynamics (captured as fibrotic area) in individual mice (*Figure 6A*). For example, leukocyte-specific integrin β2-interacting protein coronin 1a (CORO1A) displayed a significant positive correlation with fibrogenesis reflecting the time course of the immune response (*Figure 6B and C*). In contrast, negatively correlating methionine cycle enzyme adenosylhomocysteinase (AHCY) corresponded to hepatocellular death induced by fibrogenesis (*Figure 6B and C*).

In the Total CCl₄ proteome, proteins positively correlating with fibrogenesis included matrisome proteins (e.g. angiotensinogen, fibrinogen α,β,γ-chains, fibrinogen-like protein 1, and fibronectin). The IPA linked these to the inflammation and pathways associated with ECM-cell interactions, such as 'actin cytoskeleton signaling' and 'integrin signaling' (*Figure 6—figure supplement 1A*). Proteins with negative correlation were mostly enzymes involved in hepatocyte metabolism, such as alpha-enolase (ENO1), an enzyme shown to participate as plasminogen receptor in ECM degradation (*Didiasova et al., 2019*). Analogously to Total proteome, proteins of CCl₄ E-fraction positively correlating with fibrosis area were linked to fibrogenesis (*Figure 6—figure supplement 1B*). Displaying negative correlation was a group of 41 proteins consisting mainly of enzymes associated with amino acid metabolism, necrosis, and apoptosis (*Figure 6—figure supplement 1B*). In addition, three core matrisome proteins – decorin, microfibrillar-associated protein 2 (MFAP2), and collagen type I α-1 chain – were also identified. As shown for collagen type I α-1 chain, their MS intensity temporal profiles indicated their upregulation in E-fraction while their overall abundance decreased with healing (*Figure 6D*), likely corresponding to the increased association of these matrisome proteins with degraded insoluble ECM.

Among proteins positively correlating with fibrotic scar development and resolution in both Total and insoluble proteomes was clusterin (CLU), a glycoprotein implicated in several diseases (*Figure 6D*; *Wilson and Zoubeidi, 2017*). As clusterin imparts extracellular chaperone function not previously linked to fibrosis, we next investigated clusterin's localization by IF microscopy (*Figure 6E*). In untreated livers, clusterin was detected in BECs and partially in hepatocytes. Upon CCl₄ intoxication, IF staining revealed its prominent expression by damaged hepatocytes. Strikingly, clusterin increasingly re-localized along the collagen deposits over the healing period. In DDC-treated livers, clusterin was found mostly in BECs forming reactive ductuli and in hepatocytes, with minimal changes upon healing (*Figure 6E*). Moreover, extracellular clusterin deposition in close proximity of collagen fibers was further demonstrated in fibrotic and cirrhotic human livers of various etiologies at different stages of fibrosis (*Figure 6F*, *Figure 6—figure supplement 2*). The gradual increase in clusterin expression

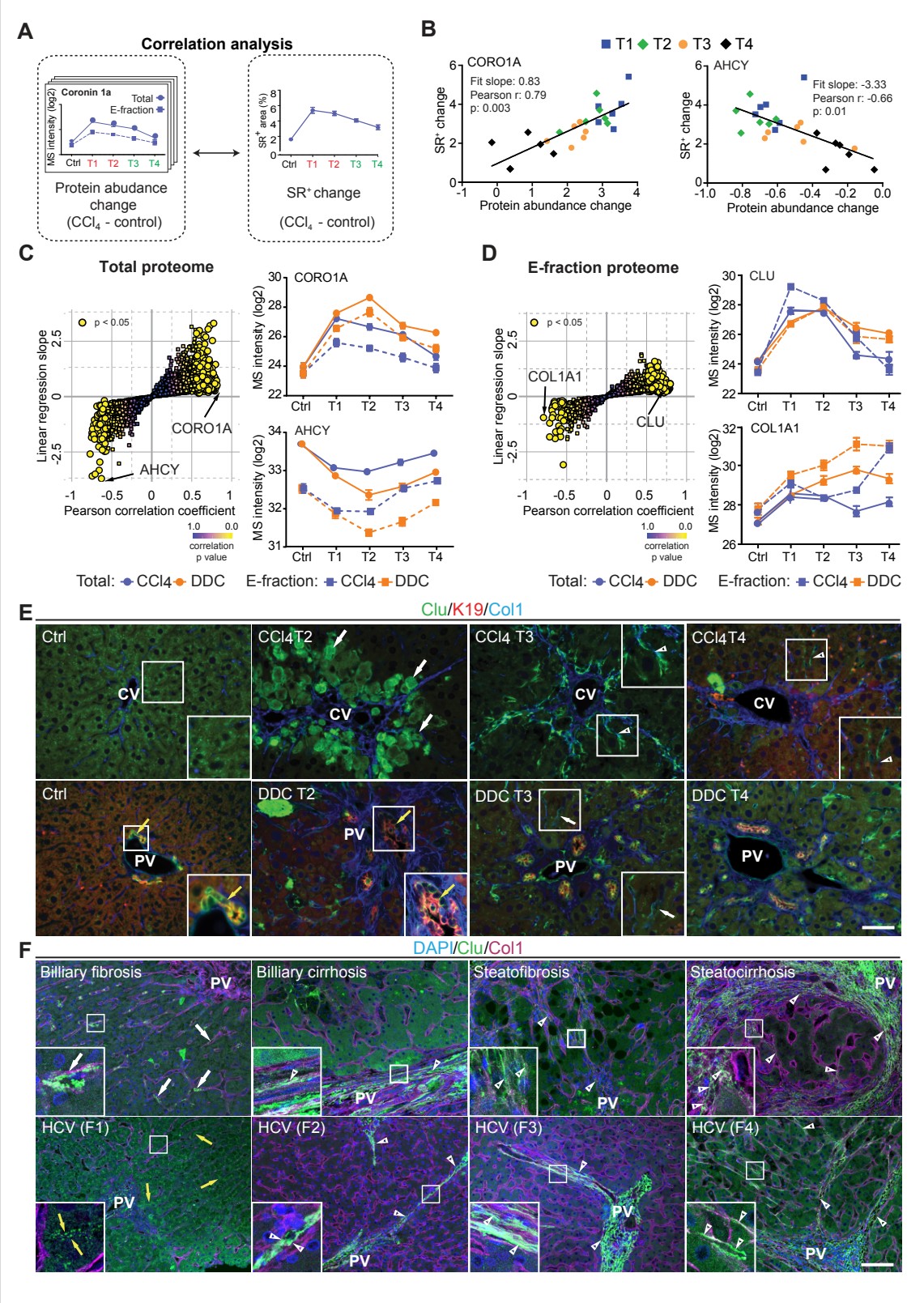

**Figure 6.** Correlation analysis of protein abundance and changes in fibrotic deposits in the hepatotoxic model associates clusterin with fibrosis resolution. (**A**) Schematic illustrates the correlation of protein abundance changes with changes in sirius red-positive (SR+) areas of fibrous extracellular matrix (ECM) deposits in carbon tetrachloride (CCl4)-treated animals at the indicated time points. (**B**) The regulator of the actin cytoskeleton, coronin 1a (CORO1A), serves as an example of a protein with a positive slope of the correlation fit. The methionine cycle enzyme, adenosylhomocysteinase

*Figure 6 continued on next page*

*Figure 6 continued*

(AHCY), serves as an example of a protein with a negative slope of the correlation fit. (**C, D**) The scatter plots show the linear regression slope and the Pearson correlation coefficient for proteins of CCl$_4$ Total (C), and E-fraction (D) proteomes. Statistical significance of the correlation is color-coded as indicated. The line plots show time-dependent change in mass spectrometry (MS) intensities of indicated representative proteins with significant correlation in Total (solid line) and E-fraction (broken line) proteomes in CCl$_4$ and 3,5-diethoxycarbonyl-1,4-dihydrocollidine (DDC) models; n=4–6. (**E**) Representative immunofluorescence (IF) images of liver sections from untreated controls (Ctrl), CCl$_4$-, and DDC-treated mice at indicated time points of fibrosis development (T2) and resolution (T3 and T4) immunolabeled for clusterin (Clu, green), keratin 19 (K19, red), and collagen type I (Col1, blue). Arrowheads, clusterin staining signal delineating collagen deposits; arrows, clusterin-positive injured hepatocytes; yellow arrows, clusterin-positive biliary epithelial cells. CV, central vein; PV, portal vein. Boxed areas, ×2 images. Scale bar = 50 µm. (**F**) Representative IF images of human liver sections from different stages of chronic liver diseases of various etiologies (biliary-type, steatotic liver disease, and chronic hepatitis C [HCV] infection) immunolabeled for clusterin (Clu, green) and collagen type I (Col1, magenta). Nuclei were stained with DAPI (blue). Top row shows increase in clusterin expression along collagen fibrils in biliary-type and metabolic syndrome-related cirrhosis compared to the stage of mild fibrosis. Bottom row documents change in clusterin staining pattern with chronic HCV progression from fibrosis stage F1 to stage F4 (METAVIR grading system: F1, portal fibrosis; F2, periportal fibrosis; F3, bridging septal fibrosis; F4, cirrhosis). Arrowheads, clusterin staining delineating collagen deposits; arrows, clusterin-positive capillarized sinusoids; yellow arrows, clusterin-positive bile canaliculi (stage F1 only). PV, portal vein. Boxed areas, ×4 images. Scale bar = 50 µm.

The online version of this article includes the following figure supplement(s) for figure 6:

**Figure supplement 1.** Ingenuity Pathway Analysis (IPA) canonical signaling pathways predicted from correlation analysis.

**Figure supplement 2.** Clusterin expression in control and diseased human livers.

**Figure supplement 3.** Correlation analysis of protein abundance and changes in fibrotic deposits in the cholestatic model.

with the progression of fibrosis against the background of chronic hepatitis C (METAVIR score F1-F4) further emphasized its role in the development of liver fibrotic diseases.

## Interface hepatocyte elasticity responds dynamically to the pericentral injury in the course of fibrosis development

Identification of 'Rho-A', 'Rho family GTPases', 'actin cytoskeleton', and 'integrin/ILK signaling' among top canonical pathways elicited by proteins of Total CCl$_4$ proteome together with the proteins of matrisome-enriched clusters (*Figure 2—figure supplement 1A and C*) suggested that the hepato-toxic injury leads to increased cytoskeletal tension in injured hepatocytes. This was further supported by significant enrichment in intermediate filament proteins (*Figure 2E*) and in MRTF targets (e.g. ITGA1, THBS1, ACTR2, and MSN; *Figure 7A*), key regulators involved in cell and tissue mechanics, cytoskeletal dynamics, and mechanosensing (*Esnault et al., 2014*).

As variations in ECM composition and content also substantially affect the biomechanics of liver tissue, we decided to examine correlation between our proteomic results and mapping of dynamic changes in the biomechanical properties of CCl$_4$-treated livers. We used atomic force microscopy (AFM) combined with polarized microscopy (*Ojha et al., 2022*) to precisely locate and probe the following compartments: (1) regions of collagen-rich scar tissue in close vicinity of the central vein, (2) injured hepatocyte regions next to the collagen scar, and (3) regions of hepatocytes on the interface between injured and not visibly damaged hepatocytes (so-called interface hepatocytes; *Ben-Moshe et al., 2022*; *Figure 7B*). In parallel, we also measured stiffness of corresponding regions in nontreated control liver sections (*Figure 7C*).

Our AFM analysis revealed no substantial topological variations across defined compartments in control livers, with median Young's moduli ranging between 1.2 and 1.6 kPa (*Figure 7C*, *Table 1*). Fibrogenic response triggered progressive tissue stiffening apparent in all analyzed compartments (*Figure 7D–F*, *Table 1*), with collagen-rich regions the stiffest at maximal fibrosis (~4.4 kPa at T2; *Figure 7D*). Unexpectedly, interface regions exhibited initial softening (T1; median ~0.8 kPa vs. 1.2 kPa in control livers), which was followed by increase in stiffness up to ~2.0 kPa (T2). With healing (T4), all compartments demonstrated clear decrease in stiffness (*Figure 7D–F*), although this was only marginal in the regions corresponding to injured hepatocytes. Significant softening of collagen-rich scar tissue during healing indicated partial resolution of scar-associated ECM while it associated with the incorporation of several core matrisome proteins (e.g. COL1A1 and FBN1) into the scar tissue (*Figures 1F and 6D*). Although the heterogeneity of AFM-based measurements constituted a limitation on our correlation analysis relative to proteomic data (*Figure 7—figure supplement 1*), this analysis provides a coherent framework for better understanding the dynamics of proteomic landscapes in the context of fibrosis-associated changes in the local mechanics of liver tissue.

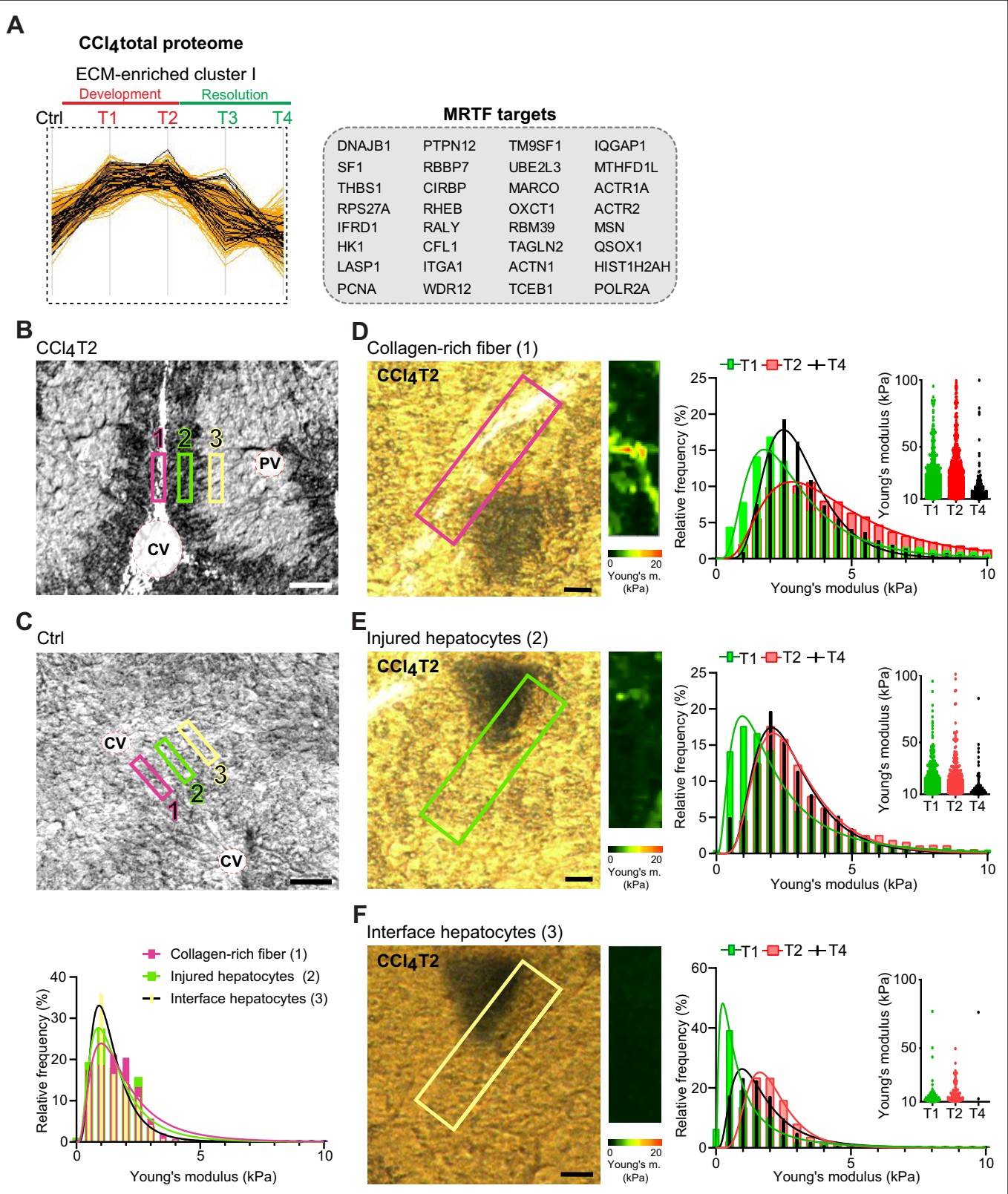

**Figure 7.** Atomic force microscopy (AFM) stiffness mapping reveals changes in the local mechanics of liver tissue upon hepatotoxic injury. (**A**) The line plot shows the dynamics of 32 myocardin-related transcription factor (MRTF) targets (highlighted in black) significantly enriched in extracellular matrix (ECM)-enriched cluster I of carbon tetrachloride (CCl₄) Total proteome (*Figure 2C*) identified by Fisher's exact test (p=0.01; Benjamini-Hochberg false discovery rate = 3%). (**B, C**) Representative polarized microscopy images of CCl₄-treated (T2, B) and untreated control (Ctrl, C) liver sections with

*Figure 7 continued on next page*

*Figure 7 continued*

indicated regions (1–3) selected for AFM measurements. Note in (B) white areas corresponding to collagen fibers visualized by polarized light. Pink rectangle, region 1 (collagen-rich fiber); green rectangle, region 2 (injured hepatocytes); yellow rectangle, region 3 (interface hepatocytes). CV, central vein; PV, portal vein. Scale bar = 100 µm. Histogram shows Young's moduli for the measured regions in untreated control livers; n=7 regions in three mice. (**D–F**) Representative polarized microscopy images of CCl$_4$-treated (T2) liver sections with rectangle indicating the regions of AFM measurements (30×100 µm$^2$) and corresponding pseudocolor Young's modulus maps determined by AFM. Scale bar = 25 µm. Histograms show Young's moduli for indicated regions of collagen-rich fibers (D), injured hepatocytes (E), and interface hepatocytes (F) at indicated time points of fibrosis development (T1 and T2) and spontaneous resolution (T4). Inset scatter plots show Young's modulus values above 10 kPa for each time point; n=7 regions in three mice.

The online version of this article includes the following figure supplement(s) for figure 7:

**Figure supplement 1.** Correlation analysis of protein abundance change and changes in mechanical properties of the local mechanics of liver tissue upon hepatotoxic injury.

## Discussion

In this study, we comprehensively characterized and compared dynamic proteomic landscapes of liver fibrosis development and repair in two mouse models based either on repetitive CCl$_4$ intoxication (*Liu et al., 2013*) or on DDC feeding (*Fickert et al., 2007*). These models are widely used to reliably mimic either hepatotoxic (CCl$_4$) or cholestatic (DDC) liver injuries leading to fibrosis in humans (*Liu et al., 2013*). Previous attempts to grasp the complexity of etiology-specific fibrotic proteomes with a focus on diseased ECM (*Massey et al., 2017*; *Klaas et al., 2016*) were limited due to origins of the analyzed material (mostly decellularized samples) and yielded only fragmented insight into the time course of the disease. To fill this research gap, we employed detergent-based tissue extraction and analyzed both Total and insoluble fraction proteomes at multiple time points of disease progression and spontaneous healing. This approach allowed us to (1) compare time-resolved, compartment-specific proteomes of CCl$_4$ and DDC models, (2) define etiology-specific elements of fibrogenesis, and (3) detect even low-abundance matrisome and matrisome-interacting proteins in the ECM-enriched insoluble fraction samples. Further, our detailed AFM-based profiling of CCl$_4$-treated livers enabled us to relate our proteomic data to disease-associated changes in the local mechanics of liver tissue.

We and others have demonstrated that the nature of liver injury determines the set of components assembled in diseased ECM (*Klaas et al., 2016*), which thus reflects unique features of the injury. For instance, the initial decline in abundance of hepatocyte- and activated BEC-expressed collagen type XVIII α-1 chain (*Schuppan et al., 1998*) and its consecutive increase thereafter in the hepatotoxic CCl$_4$ model corresponds to hepatocyte death followed by expansion of both collagen type XVIII α-1 chain-expressing cell types (*Manco et al., 2019*). By contrast, extensive collagen type XVIII α-1 chain expression in DDC-induced cholestasis was already apparent in the early fibrotic deposits due to a massive ductular reaction. A prominent group of BM-associated proteins uniquely identified in the cholestatic model can be attributed to activated BECs in proliferating bile ducts. Some of them (e.g. collagen type XVIII α-1 chain, collagen type IV and VI α-2 chains, and perlecan) are among the 14-gene signature potently predicting human patient cirrhosis progression and survival (*Wu et al., 2021*). Other DDC matrisome constituents (e.g. collagen type III α-1 chain, elastin, and laminin subunit gamma 1) have been recently linked to the deterioration of hepatocyte functions in connection with ECM stiffening (*Acun et al., 2021*). Notably, elastin is present in liver biopsies from patients with

**Table 1.** Median Young's moduli for each defined area and treatment.
Statistical analysis was performed on frequency distribution data using Kruskal-Wallis test followed by Dunn's multiple comparison post-testing. *p<0.05 , **p<0.01 vs. control.

| Treatment | Median Young's modulus (kPa) | | |
| --- | --- | --- | --- |
| | Collagen area | Affected hepatocyte area | Interface hepatocyte area |
| Control (substitute area) | 1.6 | 1.3 | 1.2 |
| 3 week CCl$_4$ (T1) | 2.7** | 1.9** | 0.8** |
| 6 week CCl$_4$ (T2) | 4.4** | 2.8** | 2.0* |
| 6 weeks+10 days (T4) | 3.0 | 2.5* | 1.4 |

advanced fibrosis and adverse clinical outcome (**Kendall et al., 2018**) and is associated with the irreversibility of liver fibrosis. Further, the increased incorporation of cross-linking protein LOXL1 and TGFβ1-related protein LTBP4 into DDC-specific ECM underscores its resemblance to human cirrhotic liver ECM (**Mazza et al., 2019**). Hence, our findings indicate that cholestasis-driven ECM deposits contain numerous proteins detected in more advanced stages of liver disease favoring hepatocarcino-genesis with a compromised ability to heal. Moreover, identified cholestasis-induced unique signaling pathways were analogous to those recently identified in a subtype of mouse cancer models mimicking human cholangiocarcinoma-like hepatocellular cancer (**Tang et al., 2022**).

Our observations are concordant with recently published studies (reviewed in **Kisseleva and Brenner, 2021**) revealing that in the fibrotic liver resident and nonresident cell types wire into dynamic intercellular hubs with shifting cell populations, reflecting and shaping the disease course in an etiology-specific manner. Many of the feedback mechanisms between the cells and ECM regulating fibrosis are mediated by members of the integrin family. In agreement with previous studies (**Nejjari et al., 2001**), we observed in both models significant upregulation of all detected integrin subunits during fibrosis progression. Strong induction of integrin αM expression in cholestatic but not hepato-toxic injury documents the role of leukocyte-specific integrin αMβ2 in the modulation of biliary fibrosis (**Joshi et al., 2016**), suggesting that the integrin αM might thus serve as a possible selective target for treatment of cholestatic liver disease. Further, a prominent transition of TGFβ1-activating integrin αv toward insoluble ECM exclusively upon hepatotoxic injury was accompanied by its localization to injured pericentral hepatocytes near collagen-rich scars and αSMA-positive HSCs. This finding, together with a measured increase in the stiffness of injured hepatocytes, implies an active role of integrin αv in the targeted activation of TGFβ1 in the local microenvironment during centrilobular fibrosis. Indeed, integrin αv located on hepatocytes in the vicinity of biliary fibrotic septa has been suggested to indicate hepatocyte biliary transformation (**Popov et al., 2008**). Here, we propose that compartment-specific induction of αv expression on the surface of scar-associated hepatocytes could be a general mechanism to promote fibrosis progression.

In contrast to the cholestatic DDC model, hepatotoxic injury in the CCl$_4$ model showed a substantial ability to heal 10 days after challenge withdrawal. Such healing capacity allowed us to correlate the dynamics in CCl$_4$ proteomes with the changes in fibrotic deposits not only during fibrosis progression but also in resolution. Among proteins correlating with scar tissue formation in both Total and insoluble proteomes, we identified a small molecular chaperone clusterin. Clusterin is believed to be associated with elastin fibers in cholestasis (**Aigelsreiter et al., 2009**), and recently it was linked to the attenuation of mouse hepatic fibrosis (**Seo et al., 2019**). Here, we localized clusterin exclusively to injured, pathologically stiffer hepatocytes. Interestingly, the clusterin promoter comprises binding motifs for mechanosensitive transcription factors such as c-Fos and AP-1/Jun (**Jin and Howe, 1999**; **Miyamoto-Sato et al., 2010**). This suggests that local clusterin induction can reflect the dynamic changes of microenvironmental mechanical cues. Most intriguingly, we also found clusterin to associate with collagen-rich ECM deposits in the course of mouse fibrosis regression and with pathological human matrix of various etiologies. As clusterin overexpression associates with increased activity of ECM-degrading matrix metalloproteinases (**Shim et al., 2011**), we hypothesize that clusterin accumulation facilitates the remodeling and resolution of scar tissue. Although the specific molecular function of clusterin in fibrotic tissue repair processes remains to be determined, our results strongly suggest clusterin to be an attractive antifibrotic target.

Liver cell and tissue mechanics play a pivotal role in the processes that initiate and resolve fibrotic injury (**Henderson et al., 2020**). Initial disruption of mechanical homeostasis prompts cytoskeletal remodeling that alters cell-generated forces and cellular biomechanics. Here, such a shift to a higher stiffness regime is illustrated by prominent changes in cytoskeleton-related signatures (actin and inter-mediate filaments) and signatures of mechanosensitive transcription regulators (e.g., MRTF) accompanied by a significant stiffening of parenchymal compartments devoid of apparent collagen deposits. This indicates substantial cell-driven changes in the biomechanical properties of tissue microenvironment. An unexpected decrease in the stiffness of interface hepatocytes with the initial fibrotic changes revealed by our AFM analysis suggests that the initial softening of interface hepatocytes upon injury counterbalances the stiffening of the injured hepatocytes and/or fibrous scar tissue regions. As the interface hepatocytes have been shown to undergo a phenotypic shift in response to the injury (**Ben-Moshe et al., 2022**), it will now be interesting to determine whether altered mechanical properties

serve as the cue leading to the genetic fetal reprogramming or if this initial softening is due to the expression of fetal markers. Together, our AFM and proteomic data underscore the role of local tissue stiffening in fibrotic response and postulate involvement of tissue softening during resolution not only in the collagen scar tissue but also in regenerating parenchyma.

# Materials and methods

**Key resources table**

| Reagent type (species) or resource | Designation | Source or reference | Identifiers | Additional information |
|---|---|---|---|---|
| Antibody | Anti- K19 (TROMA III; Rat monoclonal) | DSHB | TROMA III RRID:AB_2133570 | IHCP (1:250) |
| Antibody | Anti-αSMA (1A4; Mouse monoclonal) | DAKO | M0851 RRID:AB_2223500 | IHCP (1:50) |
| Antibody | Anti-ECM1 (F-1; Mouse monoclonal) | Santa Cruz Biotechnology | sc-365335 RRID:AB_10847810 | IHCP (1:100) |
| Antibody | Anti-clusterin (goat polyclonal) | R&D Systems | AF2747 RRID:AB_2083314 | IHCP (1:250) |
| Antibody | Anti-clusterin (A-11; Mouse monoclonal) | Santa Cruz Biotechnology | sc-166831 RRID:AB_2245186 | IHCP (1:50) |
| Antibody | Anti-fibronectin (Rabbit polyclonal) | Abcam | ab2413 RRID:AB_2262874 | IHCP (1:500) |
| Antibody | Anti-collagen I (Rabbit polyclonal) | Abcam | ab21286 RRID:AB_446161 | IHCP (1:100) |
| Antibody | Anti-collagen IV (Rabbit polyclonal) | Bio-Rad | 2150-1470 | IHCP (1:250) |
| Antibody | Anti-integrin αV(EPR16800, mouse monoclonal) | Abcam | | IHCP (1:500) |
| Antibody | Anti-B220 (RA3-6B2, mouse monoclonal) | BioLegend | 103203 | IHCP (1:100) |
| Antibody | Anti-F4/80(D2S9R mouse monoclonal) | Cell Signaling | 70076 RRID:AB_2799771 | IHCP (1:100) |
| Antibody | Anti-desmin (rabbit polyclonal) | Thermo Fisher Scientific | PA5-16705 RRID:AB_10977258 | IHCP (1:500) |
| Antibody | Rhodamine Red-X (RRX) AffiniPure Donkey Anti-Mouse IgG (H+L) | Jackson ImmunoResearch | 715-295-151 RRID:AB_2340832 | IHCP (1:250) |
| Antibody | Alexa Fluor 488 AffiniPure F(ab')₂ Fragment Donkey Anti-Goat IgG (H+L) | Jackson ImmunoResearch | 705-546-147 RRID:AB_2340430 | IHCP (1:250) |
| Antibody | Alexa Fluor 594 AffiniPure F(ab')₂ Fragment Donkey Anti-Rat IgG (H+L) | Jackson ImmunoResearch | 712-586-150 RRID:AB_2340690 | IHCP (1:250) |
| Antibody | Alexa Fluor 647 AffiniPure Donkey Anti-Rabbit IgG (H+L) | Jackson ImmunoResearch | 711-605-152 RRID:AB_2492288 | IHCP (1:250) |
| Antibody | Alexa Fluor 488 AffiniPure Donkey Anti-Mouse IgG (H+L) | Jackson ImmunoResearch | 715-545-150 RRID:AB_2340846 | IHCP (1:250) |
| Antibody | Alexa Fluor 488 AffiniPure Donkey Anti-Rabbit IgG (H+L) | Jackson ImmunoResearch | 711-545-152 RRID:AB_2313584 | IHCP (1:250) |
| Chemical compound, drug | CCl4 | Sigma-Aldrich | SML1656 | |
| Software, algorithm | SPSS | SPSS | RRID:SCR_002865 | |
| Other | DAPI stain | Invitrogen | D1306 | (1 µg/ml) |
| Software | Perseus | https://maxquant.net/perseus/ | | Version 1.6.10.43 |

## Ethical statement

### Human samples

The use of completely anonymized archived liver tissue samples for research purposes has been approved by the Ethical Committee of the Institute of Experimental and Clinical Medicine and Thomayer University Hospital, Prague, Czech Republic. Written informed consent was obtained from all patients enrolled in the study. All research was conducted in accordance with both the Declarations of Helsinki and Istanbul.

### Animals

Male C57BL/6J mice (8–10 weeks of age) were housed under standard pathogen-free conditions with free access to regular chow and drinking water and a 12-hr-dark/12-hr-light cycle. All animal experiments were performed in accordance with an animal protocol approved by the Animal Care Committee of The Institute of Molecular Genetics, Prague, Czech Republic and according to EU Directive 2010/63/EU for animal experiments.

### Liver injury models

To induce fibrogenesis, mice were either injected intraperitoneally with 1 µl/g $CCl_4$ (diluted 1:3 with olive oil) twice a week for 3 or 6 weeks or fed a diet supplemented with 0.1% DDC (Sniff, Soest, Germany) for 2 or 4 weeks. To study fibrosis resolution, mice treated with $CCl_4$ for 6 weeks or fed DDC for 4 weeks were removed from treatment and allowed to recover for 5–10 days. At the end of treatment, mice were anesthetized, blood was collected into heparin-containing tubes (KabeLabortechnik, Numbrecht, Germany) to obtain plasma, and mice were sacrificed by cervical dislocation to collect liver samples. Plasma levels of aspartate transaminase, alanine transaminase, alkaline phosphatase, and total bilirubin were measured using commercial kits (Roche Diagnostics, Prague, Czech Republic).

A 1-mm-thick middle section of left lateral liver lobe was removed and two ~1 $mm^3$ pieces were cut, flash-frozen in liquid nitrogen, and stored at –80°C for further processing for mass spectrometry (MS). The rest of the lobe was divided into two parts: one was fixed in 4% buffered formalin (pH 7.4) followed by paraffin embedding for histology and immunostaining and the other was embedded in OCT Tissue-Tek (Sakura Finetek, USA) for further cryosectioning for AFM measurements.

## Histology and IF

Formalin-fixed, paraffin-embedded liver sections (4 µm; both mouse and human) were stained with hematoxylin and eosin and sirius red. For IF, these sections were de-paraffinized and subjected to heat-induced antigen retrieval in Tris-EDTA (pH 9) or citrate (pH 6) buffer and further permeabilized with 0.1 M glycine, 0.1% Triton X-100 (15 min). Next, sections were incubated with primary antibodies overnight at 4°C, followed by incubation with secondary antibodies for 120 min at 22°C. The following primary antibodies were used: rat mAb to K19 (Troma III; Developmental Studies Hybridoma Bank, University of Iowa, Iowa City, IA, USA); mouse mAbs to αSMA (DAKO, Glostrup, Denmark) and ECM1 (Santa Cruz Biotechnology, Dallas, TX, USA), goat polyclonal antibodies to clusterin (R&D Systems, Minneapolis, MN, USA), rabbit polyclonal antibodies to fibronectin, collagen type I, collagen type IV, and monoclonal to integrin αV (all from Abcam, Cambridge, UK). As secondary antibodies we used donkey anti-mouse IgG Alexa Fluor (AF) 488 and 647, goat anti-mouse IgG and goat anti-rabbit IgG horseradish peroxidase-conjugated, donkey anti-rabbit IgG AF488, donkey anti-rat IgG AF594 (all from Jackson ImmunoResearch, Baltimore, MD, USA). ECM1 and integrin αV were visualized using Alexa Fluor 488 Tyramide Reagent (Thermo Fisher Scientific, Prague, Czech Republic) according to the manufacturer's instructions. Stained sections were further mounted with ProLongGold Antifade mounting media (Thermo Fisher Scientific) and images were acquired with inverted fluorescence microscope DM6000 (Leica Microsystems, Pragolab, Prague, Czech Republic) equipped with HC PL Apo ×10/0.40 NA, PH1 HC PLAN Apo ×20/0.70 NA, and PH2 HCX PL Apo ×40/0.75 NA, objective lenses.

## MS sample preparation

A sample of ca 1 $mm^3$ excised from the middle section of left lateral liver lobe was cryo-homogenized using TissueLyser II (QIAGEN, Germantown, MD, USA). The cryo-homogenized liver powder was then

resuspended in 200 µl of 50 mM Tris-HCl pH 7.5, 5% glycerol, 500 mM NaCl, 1% IGEPAL CA-630, 2% sodium deoxycholate, 1% SDS, 1× Proteinase Inhibitor Cocktail. The suspension was sonicated after 20 min incubation at 22°C. A 50 µl aliquot of the resuspended sample was further precipitated with ice-cold acetone and labeled 'Total'. The remaining solution was centrifuged at 16,000 × g for 20 min to separate the supernatant containing soluble (S)-fraction from the pellet, which itself contained insoluble ECM-enriched (E)-fraction. Supernatants were further precipitated with ice-cold acetone and stored together with pellets at –80°C for further processing preceding MS analysis.

## Protein digestion

Protein pellets were homogenized and lysed in 100 mM triethylammonium bicarbonate (TEAB) containing 2% sodium deoxycholate, 40 mM chloroacetamide, 10 mM Tris(2-carboxyethyl)phosphine at 95°C for 10 min followed by sonication (Bandelin Sonopuls Mini 20 homogenizer). Protein concentration was determined using a BCA protein assay kit (Thermo Fisher Scientific) and 30 µg of protein per sample was used for MS sample preparation as described before (*Hughes et al., 2019*). Briefly, 5 µl of SP3 beads were added to 30 µg of protein sample and brought to 50 µl with 100 mM TEAB. Protein binding was induced by the addition of ethanol to 60% (vol/vol) final concentration. Samples were mixed and incubated for 5 min at 22°C. After binding, tubes were placed into the magnetic rack and the unbound supernatant was discarded. Beads were subsequently washed two times with 180 µl of 80% ethanol. Samples were further digested with trypsin (trypsin/protein ratio 1/30, reconstituted in 100 mM TEAB) at 37°C overnight followed by acidification with trifluoroacetic acid to 1% final concentration. Peptides were desalted using in-house-made stage tips packed with 3M Empore C8 extraction disks (Thermo Fisher Scientific) as described before (*Rappsilber et al., 2007*).

## nLC-MS/MS analysis

Nano Reversed phase columns (EASY-Spray column, 50 cm × 75 µm ID, PepMap C18, 2 µm particles, 100 Å pore size) were used for LC/MS analysis using 0.1% formic acid in water as mobile phase buffer A and 0.1% formic acid in acetonitrile as mobile phase B. Samples were loaded onto the trap column (C18 PepMap100, 5 µm particle size, 300 µm × 5 mm; Thermo Fisher Scientific) for 4 min at 18 µl/min in loading buffer (2% acetonitrile, 0.1% trifluoroacetic acid). Peptides were eluted with mobile phase B gradient from 4% to 35% in 120 min. Eluting peptide cations were converted to gas-phase ions by electrospray ionization and analyzed on a Thermo Orbitrap Fusion (Q-OT-qIT, Thermo Scientific). Survey scans of peptide precursors from 350 to 1400 m/z were performed in orbitrap at 120 K resolution (at 200 m/z) with a $5 \times 10^5$ ion count target. Tandem MS was performed by isolation at 1.5 Th with the quadrupole, HCD fragmentation with a normalized collision energy of 30, and rapid scan MS analysis in the ion trap. The MS2 ion count target was set to $10^4$ with the max injection time of 35 ms. Only those precursors with charge state 2–6 were sampled for MS2. The dynamic exclusion duration was set to 45 s with a 10 ppm tolerance around the selected precursor and its isotopes. Monoisotopic precursor selection was turned on. The instrument was run in top speed mode with 2 s cycles (*Hebert et al., 2014*).

## MS intensity data quantification

Peptides were identified and quantified by MaxQuant label-free quantification software (1.6.7 version; *Cox and Mann, 2008*) using *Mus musculus* UniProt protein database (UniProtKB version July 2020, containing 25,805 entries) and the instrument type set to Orbitrap. Trypsin was set as the proteolytic enzyme and two missed cleavages were allowed. Carbamidomethylation of cysteine was selected as a fixed modification and N-terminal protein acetylation and methionine oxidation as variable modifications. Reverse sequences were selected for the target-decoy database strategy (*Elias and Gygi, 2010*). The FDR was set to 1% for both proteins and peptides, a minimum peptide length specified for seven amino acids. The 'match between runs' feature of MaxQuant was used to transfer identifications to other LC-MS/MS runs based on their masses and retention time and this was also used in quantification experiments. Quantifications were performed with the label-free algorithm in MaxQuant (*Cox et al., 2014*). The Fast LFQ option was switched off. Each sample of total protein extract was treated as a separate 'experiment' for quantification. Corresponding E- and S-fractions together were treated as another separate 'experiment'. Unique+razor values were used for label-free quantification (LFQ). Protein groups identified through target-decoy database strategy ('Reverse' column in the

MaxQuantproteinGroups file) and proteins not identified by MS/MS spectrum ('Only identified by site' column in the MaxQuantproteinGroups file) were removed and not included in any subsequent analysis.

The overall protein intensity for total extract samples was estimated as a sum of MaxQuant LFQ values for unique and razor-detected peptides. For each S-fraction sample and corresponding E-fraction sample, the proportional coefficients for LFQ values of unique and razor-detected peptides of each 'experiment' were computed according to *Cox et al., 2014*, using the script for R software (version 4.0.5 for Windows) and the values in the MaxQuant evidence file. After the estimation of the proportional coefficients for each fraction, the overall protein MS intensity for each fraction was estimated as a sum of normalized LFQ values for unique and razor-detected peptides. The MS proteomics data have been deposited to the ProteomeXchange Consortium via the PRIDE (*Perez-Riverol et al., 2025*) partner repository with the dataset identifier PXD060305 and are accessible with token 65Dd5LEEHtWr.

## Bioinformatic analysis and statistics

Normalized (*Hynek et al., 2021*; *Cejnar et al., 2022*) log 2-transformed MS intensity data quantified as described above were uploaded for bioinformatic analysis into the bioinformatics platform Perseus (1.6.10.43 version; *Tyanova et al., 2016*). The used annotations were: gene ontology, biological process, molecular function, cellular component, KEGG pathway databases, matrisome database (*Naba et al., 2012*; *Shao et al., 2020*), previously published cell-type-specific protein and gene expression signatures of liver cells (*Azimifar et al., 2014*) and leukocyte populations (*Gautier et al., 2012*; *Miller et al., 2012*; *Jojic et al., 2013*), annotations, 'location', and 'function' from IPA (*Krämer et al., 2014*) annotation (IPA, QIAGEN Inc, https://www.qiagenbioinformatics.com/products/ingenuitypathway) and MRTF target genes (*Esnault et al., 2014*). All data analyzed were first log 2-transformed. The PCA was performed in Perseus software with the built-in tool and all data imputed by random selection from a normal distribution generated at 1.8 standard deviations subtracted from the mean of the total intensity distribution and a width of 0.3 standard deviations. For Total proteome sample analysis, data from Total samples were first filtered for proteins expressed in at least one group (control, and times T1, T2, T3, or T4) with at least five identified values in both models ($CCl_4$ and DDC) together. For subsequent analyses we also similarly filtered Total proteome samples in each model separately. Missing values among identified proteins in analyzed datasets were imputed into Perseus as described above. Proteins significantly differentially expressed in the Total proteome of each fibrosis model separately (analyses in *Figures 2 and 3*) were defined as proteins with a minimum 1.5-fold significant change in expression in the highest fibrosis time point (T2) over the average expression in controls (t-test at a BH FDR<0.05). Unsupervised hierarchical cluster analyses of averages of z-scored MS intensities of identified proteins were performed with Euclidean distance and complete linkage in the Perseus software with built-in cluster enrichment analysis. To extract individual cell-type dynamics from the proteomics data, we used previously published cell-type-specific signatures (*Azimifar et al., 2014*; *Gautier et al., 2012*; *Miller et al., 2012*; *Jojic et al., 2013*). For each of this way identified proteins, average z-scored MS intensity in each time point was calculated and the signature proteins of each cell type were together plotted in each time point as box plots, with 25th and 75th percentile indicated as explained below.

All graphs and statistical tests indicated in graphs were performed using GraphPad Prism version 9.3.1 (GraphPad Software, Boston, MA, USA, https://www.graphpad.com/). All results are presented as mean ± SEM. In the box plots, the box represents the 25th and 75th percentile with the median indicated; whiskers reach the last data point; dots indicate means of independent experiments. Normally distributed parametrical data were analyzed by two-tailed unpaired Student's t-test. A comparison between multiple groups was performed using one-way ANOVA with Tukey's multiple comparison test. Data distribution was assumed to be normal, but this was not formally tested. Statistical tests used are specified in the figure legends. Statistical significance was determined at the level of *, $p < 0.05$; **, $p < 0.01$; [†], $p < 0.001$.

## E-fraction analysis

Proteins identified in E-fraction samples were first filtered and imputed as described above for each model separately. All proteins detected in S-fraction were also imputed in Perseus as described above

and median S-fraction MS intensity value for each protein was calculated. To identify proteins with change in solubility, which would be specifically enriched in E-fraction, we calculated a ratio of the MS intensity of each protein replicate in E-fraction to the median of MS intensity of this protein in S-fraction and selected those with a minimum threefold higher expression in E- than S-fraction in at least one time point (control, T1-T4). From these, proteins with a significant change in time identified by ANOVA with BH FDR<0.05 represented insoluble fraction proteins, i.e., E-fraction proteome. Unsupervised hierarchical clustering of MS intensity ratios of the abundance of these proteins in E- to S-fraction was performed in Perseus as described above to identify changes in protein solubility over time with disease progression and recovery.

### Ingenuity Pathway Analysis

We used the IPA (*Krämer et al., 2014*) to identify predicted upstream transcriptional regulators and growth factors and downstream canonical signaling pathways elicited by groups of proteins identified in individual analyses. Transcriptional regulators and growth factors were identified in upstream regulator analysis with protein overlap p-value>3 (log 10) and activation z-score>|2.5|. Canonical pathways were considered only with overlap p-value>3 (log 10) and activation z-score>|2|.

### Correlation analysis

The MS protein temporal abundance change profile of each protein was correlated with the fibrosis area change for proteins of Total proteome and E-fraction proteome in each model. To do this, we calculated the mean value of log 2-transformed and imputed MS intensities for each protein in controls and subtracted it from the log 2-transformed MS intensity for each protein at each time point in individual mice. This way obtained temporal abundance profile ratio of each protein was then correlated to the temporal profile of fibrosis area (calculated as a percentage of sirius red positive areas in each mouse at each time point after subtracting the background calculated as the average sirius red positive area in control livers) using Pearson correlation coefficient separately for each model. We then plotted the correlation linear regression slope against the Pearson correlation coefficient in each of the proteomes for each of the identified proteins (*Figure 6B–D*). The significance of dependency (slope's p-value) for each protein was corrected by BH multiple comparisons test (p = 0.05). Analysis of DDC-induced differentially expressed proteins did not result in any proteins significantly correlating with collagen deposition in either of the proteomes (*Figure 6—figure supplement 3*). This is most likely a consequence of considerable variability in the scar tissue deposition in this model at each time point.

### Atomic force microscopy

AFM was performed on 30-μm-thick liver sections shortly fixed with 4% PFA in PBS, as previously described (*Ojha et al., 2022*). Briefly, glass slides containing fixed liver sections covered with PBS were placed onto a combined AFM/inverted optical microscope (JPK NanoWizard 3XP; Bruker Nano Nano Surfaces Division, Santa Barbara, CA, USA/Olympus IX81; Olympus C&S, Prague, Czech Republic) equipped with a custom-made dual-polarization motorized filter allowing precise localization areas of interest: collagen-rich fibrotic areas, areas of injured, and interface hepatocytes (*Ben-Moshe et al., 2022*). To characterize the mechanical properties of livers, we used an SD-qp-BioT-TL-10 cantilever (S/N-73750F05, Nanosensors, NanoAndMore GmbH, Wetzlar, Germany) with a nominal sprint constant of 0.09 N/m modified with a 5.7 μm melamine polymer bead (Microparticles GmbH, Germany). Before each experiment, the cantilever spring constant was calibrated using the thermal noise method while immersed in PBS. Using the force mapping method, we measured 30 ×100 μm$^2$ (10 × 36 pixels) defined areas precisely located by polarized microscopy. We analyzed seven areas (measured from two sections per mouse) obtained from three different mice for each time point (control, T1, T2, and T4). Data were further processed with open-source software 'AtomicJ' (*Hermanowicz et al., 2014*) using a Poisson ratio of 0.45 (*Barnes et al., 2007*). All data were further analyzed in GraphPad Prism software and are presented as frequency distribution histograms. Statistical analyses were performed on frequency distribution histograms using the Kruskal-Wallis test followed by Dunn's multiple comparisons post-testing. To correlate the AFM data with protein MS abundance change during fibrosis development and resolution, we averaged median Young's moduli for each analyzed area in each measured mouse. After subtraction of mean control Young's moduli, the slope

and p-value for the dependency of the stiffness of the area on the protein MS intensities were determined, and the p-value was corrected by BH multiple comparisons test (p<0.05).

## Acknowledgements

We would like to thank SM Meier-Menches (University of Vienna, Austria) and J Masek (Charles University, Prague, Czech Republic) for critical reading of the manuscript and D Hadraba (Institute of Physiology, Prague, Czech Republic) for sharing his expertise. We further acknowledge the Light Microscopy Core Facility, IMG CAS, Prague, Czech Republic for support with the microscopy imaging presented herein. We would also like to thank V Getmanchuk – Zaporoshchenko for detailed proof-reading of the final manuscript. This work was supported by the Grant Agency of the Czech Republic (18-02699S and 21-21736S); the Grant Agency of the Ministry of Health of the Czech Republic (NU21-04-00100); the Institutional Research Project of the Czech Academy of Sciences (RVO 68378050); the National Institute for Cancer Research (Program EXCELES, LX22NPO5102) – funded by the European Union – Next Generation EU; the Grant Agency of Charles University (273723), and MEYS CR projects (LM2023050, LM2018126, LQ1604 NPU II, LO1419, and LM2015040). We acknowledge CF Nanobiotechnology of CIISB, Instruct-CZ Centre, supported by MEYS CR (LM2023042) and European Regional Development Fund-Project 'Innovation of Czech Infrastructure for Integrative Structural Biology' (No. CZ.02.01.01/00/23_015/0008175). The funding sources were not involved in the study design, data collection and analysis, decision to publish, or preparation of the article.

## Additional information

### Funding

| Funder | Grant reference number | Author |
|---|---|---|
| Grantová Agentura České Republiky | 18-02699S | Marketa Jirouskova |
| Grantová Agentura České Republiky | 21-21736S | Martin Gregor |
| Ministerstvo Školství, Mládeže a Tělovýchovy | NU21-04-00100 | Martin Gregor |
| Ministerstvo Školství, Mládeže a Tělovýchovy | RVO 68378050 | Martin Gregor |
| NextGenerationEU | LX22NPO5102 | Martin Gregor |
| Grantová Agentura, Univerzita Karlova | 273723 | Srikant Ojha |
| Ministerstvo Školství, Mládeže a Tělovýchovy | LM2023050 | Martin Gregor |
| Ministerstvo Školství, Mládeže a Tělovýchovy | LM2018126 | Martin Gregor |
| Ministerstvo Školství, Mládeže a Tělovýchovy | LQ1604 NPU II | Martin Gregor |
| Ministerstvo Školství, Mládeže a Tělovýchovy | LO1419 | Martin Gregor |
| Ministerstvo Školství, Mládeže a Tělovýchovy | LM2015040 | Martin Gregor |

The funders had no role in study design, data collection and interpretation, or the decision to submit the work for publication.

### Author contributions

Marketa Jirouskova, Conceptualization, Data curation, Formal analysis, Funding acquisition, Investigation, Writing – original draft, Writing – review and editing; Karel Harant, Data curation, Formal

analysis, Methodology, Writing – review and editing; Pavel Cejnar, Data curation, Formal analysis, Writing – review and editing; Srikant Ojha, Data curation, Formal analysis, Funding acquisition, Investigation, Writing – review and editing; Katerina Korelova, Visualization, Project administration, Writing – review and editing; Lenka Sarnova, Data curation, Investigation; Eva Sticova, Christoph H Mayr, Herbert B Schiller, Resources, Writing – review and editing; Martin Gregor, Conceptualization, Supervision, Funding acquisition, Writing – original draft, Writing – review and editing

### Author ORCIDs

Marketa Jirouskova (ID) https://orcid.org/0009-0004-7173-2177
Martin Gregor (ID) https://orcid.org/0000-0001-6841-9527

### Ethics

The use of completely anonymized archived liver tissue samples for research purposes has been approved by the Ethical Committee of the Institute of Experimental and Clinical Medicine and Thomayer University Hospital, Prague, Czech Republic. Written informed consent was obtained from all patients enrolled in the study. All research was conducted in accordance with both the Declarations of Helsinki and Istanbul.

All animal experiments were performed in accordance with an animal protocol approved by the Animal Care Committee of The Institute of Molecular Genetics and according to EU Directive 2010/63/EU for animal experiments.

Reviewer #1 (Public review): https://doi.org/10.7554/eLife.98023.3.sa1
Reviewer #2 (Public review): https://doi.org/10.7554/eLife.98023.3.sa2
Author response https://doi.org/10.7554/eLife.98023.3.sa3

---

# Additional files

### Supplementary files

MDAR checklist

### Data availability

The mass spectrometry proteomics data have been deposited to the ProteomeXchange Consortium via the PRIDE partner repository (*Perez-Riverol et al., 2025*) with the dataset identifier PXD060305.

The following dataset was generated:

| Author(s) | Year | Dataset title | Dataset URL | Database and Identifier |
|---|---|---|---|---|
| Jirouskova M, Harant K, Cejnar P, Ojha S, Korelova K, Sarnova L, Sticova E, Mayr CH, Schiller HB, Gregor M | 2025 | Dynamics of compartment-specific proteomic landscapes of hepatotoxic and cholestatic models of liver fibrosis | https://www.ebi.ac.uk/pride/archive/projects/PXD060305 | PRIDE, PXD060305 |

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
