## [Editor Report · eLife Assessment]

This **important** study suggests that the composition of the extracellular matrix in a mouse model of liver fibrosis changes depending on the cause of liver fibrosis. The data could be used as a foundation for future antifibrotic therapies. The strength of evidence is **convincing** with respect to the use of animal models and proteomic analysis. The study provides a helpful inventory of proteins up or down-regulated.

---

## [Referee Report · Reviewer #1 (Public review)]

Summary:

Jirouskova and colleagues in their study have carried out an in depth proteomic characterization of the dynamics of the liver fibrotic response and the resulting resolution in two distinct models of liver injury: CCl4-induced model of hepatotoxicity and pericentral/bridging liver fibrosis and the DDC feeding model of obstructive cholestasis and periportal fibrosis. They focussed on both the insoluble extracellular matrix (ECM) components as well as the soluble secreted factors produced by hepatic stellate cells (HSCs) and/or portal fibroblasts (PFs). They identified compartment- and time-resolved proteomic signatures in the two models with disease-specific factors or matrisomes. Their study also identified phenotypic differences between the models such as that while the CCl4-induced model induced profound hepatotoxicity followed by resolution, the DDC model induced more lasting liver damage and proteomic changes that resembled advanced human liver fibrosis favouring hepatocarcinogenesis.

Overall, this comprehensive and very well conducted study is rigorous and well planned. The conclusions are supported by compelling studies and analyses. One caveat is the lack of mechanistic experiments to prove causality, but this can be carried out in follow-up studies.

Strengths:

• A major strength in the study is that the experiments are rigorous and very well conducted. For instance, the authors utilized two models of liver fibrosis to study different aspects of the pathology - hepatotoxicity vs cholestasis. In addition, 4 time points for each model were investigated - 2 for fibrosis development and 2 for fibrosis resolution. They have taken 3 components for proteomic analyses - total lysates, insoluble ECM components as well as the soluble secreted factors. Thus, the authors provide a comprehensive overview of the fibrosis and resolution process in these models.

• Another great strength of the study is that the methodology utilized was able to dissect unique pathways relevant for each model as well as common targets. For example, the authors identified known pathways such as mTOR signalling to be differentially regulated in the CCl4 vs DDC model. mTOR signalling was increased in the DDC model that is associated with hyperproliferation. Thus showing that the approach taken is specific enough to distinguish between the two similar (both induce fibrosis) but distinct mechanisms (hepatotoxicity vs cholestasis) is a strong point of the study.

Weaknesses:

• A caveat of the study is that the authors have not conducted mechanistic (gain of function/loss of function) studies from any of their identified targets to truly prove causality. This remains one of the limitations of this study. Thus, future studies should investigate this point in detail. For instance, it would have been intriguing to dissect if knocking out specific genes involved in one specific model or genes common to both would yield distinct phenotypic outcomes.

---

## [Referee Report · Reviewer #2 (Public review)]

Summary:

The authors suggest that ECM abundance and composition change depending on the aetiology of liver fibrosis. To understand this they have investigated the proteome in two models of animal fibrosis and resolution. They suggest their findings could provide a foundation for future anti-fibrotic therapies.

The revised version has been improved. Although some areas remain (described below), it is perhaps the dataset that will be most valuable.

Strengths:

The dataset appears well supported and will be valuable.

Weaknesses:

The manuscript is still fairly descriptive but on balance this is a useful dataset and appears to have broad support in that regard.

There are no conclusions that can be drawn from their rebuttal regarding the human data they included as it is one patient per group and will most likely change dramatically with more patients. As such this area is still an issue but they have improved some of the data elsewhere.

---

## [Author Response]

The following is the authors’ response to the original reviews.

**Reviewer #1:**
Weaknesses:(1) The authors themselves propose in their Introduction that the "ECM-associated changes are increasingly perceived as causative, rather than consequential"; however, they have not conducted mechanistic (gain of function/loss of function) studies either in vitro or in vivo from any of their identified targets to truly prove causality. This remains one of the limitations of this study. Thus, future studies should investigate this point in detail. For instance, it would have been intriguing to dissect if knocking out specific genes involved in one specific model or genes common to both would yield distinct phenotypic outcomes.

We agree with the reviewer that our study does not provide mechanistic verification of the function of identified targets with suggested role in the development and/or resolution of fibrosis. The current study was primarily conducted in order to identify these possible targets with focus on the identification of differences in extracellular matrix deposited in two selected models of liver fibrosis with different modes of action. To conduct further studies using knock-out/in models for verification of causality of proposed targets was at this point well beyond our intention. However, we are fully aware of the potential of identified molecules and further studies to disect their roles in liver diseases are part of future plans.

(2) The majority of the conclusions are derived primarily from the proteomic analyses. Although well conducted, it would strengthen the study to corroborate some of the major findings by other means such as IHC/IF with the corresponding quantifications and not only representative images.

We have now provided additional IF images and their quantifications in accordance with the Reviewer’s suggestions to our major MS findings to strenghten the significance of the MS data (see detailed answer below).

**Reviewer #2:**
Weaknesses:(1) As it currently stands, the data, whilst extensive, is primarily focussed on the proteomic data which is fairly descriptive and I am not clear on the additional insight gained in their approach that is not already detailed from the extensive transcriptomic studies. The manuscript overall would benefit from some mechanistic functional insight to provide new additional modes of action relevant to fibrosis progression.

We agree with the reviewer that our study could initially appear descriptive. However, this characteristics is inherent to most omics studies, which tend to provide hypothesis-free testing of a large number of analytes in order to find a multitude of candidate biomarkers(1). Importantly, we believe our study provides insights that go beyond the scope of previously published transcriptomic analyses.

Specifically, our work focuses on compartment-specific changes in the liver proteome, with an emphasis on the extracellular matrix (ECM) composition and alterations in protein solubility—features that cannot be captured by transcriptomic studies. The matrisome is more than a structural scaffold; it functions as a reservoir for secreted factors, including growth factors and cytokines, which modulate the local cellular microenvironment. Transition dynamics between the insoluble matrisome and soluble protein pools influence the signaling capabilities and bioavailability of these factors. Moreover, fibrous ECM assemblies directly impact tissue mechanics, providing cells embedded within the matrix with spatially distinct biochemical and biomechanical contexts. The current understanding of matrisome composition in the context of specific liver disease etiologies is limited. Dr. Friedman, in his 2022 review on hepatic fibrosis, highlights the unmet need to elucidate etiology-specific protein signatures of the cirrhotic liver matrisome, which could serve as disease staging or prognostic biomarkers(2). Our study addresses this gap by characterizing the distinct matrisome profiles associated with hepatotoxic- versus cholestasis-driven liver injury. We believe our findings lay the groundwork for identifying etiology-specific biomarkers and potential therapeutic targets for antifibrotic interventions, offering a novel layer of insight beyond what transcriptomic data alone can provide.

(2) Whilst there is some human data presented it is a minimal analysis without quantification that would imply relevance to disease state. Although studying disease progression in animals is a fundamental aspect of understanding the full physiological response of fibrotic disease, without more human insight makes any analysis difficult to fulfil their suggestion that these targets identified will be of use to treat human disease.

We thank the reviewer for this comment. Our study primarily focuses on utilizing animal models to explore the fundamental physiological processes underlying the development and resolution of fibrotic liver disease. To address the translational relevance of our findings, we concentrated on clusterin, one of the key target proteins identified during our analysis of the insoluble proteome. Specifically, we investigated its localization in human liver samples, focusing on its association with collagen deposits (Figure 6F). To this end, we analyzed human liver samples of diverse etiologies and varying degrees of fibrotic damage, including samples representing four distinct stages of HCV-induced fibrosis (Figure 6F, lower panel). While this analysis highlights the presence and localization of clusterin in fibrotic deposits, we acknowledge that our study does not include extensive quantification or mechanistic insight into clusterin's role in human liver fibrosis. We believe that the data presented in this manuscript provide a valuable foundation for future investigations into clusterin’s involvement in liver fibrosis across different etiologies. Recognizing the translational importance of this work, we have already initiated a prospective study involving human patients, which aims to conduct a more comprehensive analysis of clusterin's function and its potential as a therapeutic target.

To further support our findings on clusterin's role in fibrosis development and resolution and to address the reviewer's concern, we quantified clusterin deposits in the available human samples representing four distinct stages of HCV-induced fibrotic disease. Using immunofluorescence (IF) images at a 20x field of view, we measured both clusterin and collagen deposits to illustrate changes in clusterin abundance during fibrosis progression (stages F1–F4) in relation to collagen deposition dynamics. The quantified data have been included for the reviewer's consideration (Figure 1). However, it is important to emphasize that this quantification was conducted on a single human sample per fibrotic stage, which limits the statistical robustness of the analysis. A more comprehensive evaluation involving additional patient samples would be necessary for a more definitive conclusion. For this reason, we propose to include these results solely in our rebuttal letter and to incorporate a more extensive analysis in our intended follow-up study, where larger cohorts will allow for a thorough investigation of clusterin's role in human liver fibrosis.

**Author response image 1. sa3fig1:** Dynamics of clusterin abundance with the development of HCV-induced fibrotic disease in comparison to the changes in collagen deposits. IF images of human liver sections from different stages of chronic HCV infection were immunolabeled for clusterin and collagen 1. Clusterin- and collagenpositive (^+^) areas (as %) from three to eight fields of view (20x objective) were evaluated for each fibrosis stage (F1-F4).

(3) Some of the terminology is incorrect while discussing these models of injury used and care should be taken. For example - both models are toxin-induced and I do not think these data have any support that the DDC model has a higher carcinogenic risk. An investigation into the tumour-induced risk would require significant additional models. These types of statements are incorrect and not supported by this study.

We are grateful to the reviewer for drawing our attention to the incorrect use of the term "toxin-induced". In two instances, where the wording was incorrect, we have corrected the term to hepatotoxin-induced as it was originally intended. While we believe that our proteomic signature data and identified signaling pathways suggest a potential carcinogenic risk associated with the cholestatic, but not the hepatotoxic model, we have toned down the statements on this issue in the article to respect the reviewer's perspective. These changes, which are highlighted in the track changes mode of the article, aim to make the conclusions of the study more precise and thus improve the clarity of our conclusions.

**Reviewer #1 (Recommendations for the authors):**
(1) In the Discussion, the authors could consider pointing out that one limitation of the study is a lack of mechanistic (gain of function/loss of function) studies either in vitro or in vivo from any of their identified targets to truly prove causality.

As noted earlier, we fully agree with both reviewers that a limitation of this study is its descriptive nature, which is an inherent characteristic of omics-based research. In our manuscript, we aimed to "determine compartment-specific proteomic landscapes of liver fibrosis and delineate etiology-specific ECM components," with the overarching goal of providing a foundation for future antifibrotic therapies.

The insights gained from our study will indeed serve as a critical basis for subsequent research, where we will prioritize mechanistic investigations to elucidate the roles of the identified targets. While we acknowledge the importance of gain- or loss-of-function studies to establish causality, we believe this falls outside the primary scope of the current manuscript. Instead, we envision these mechanistic approaches as key elements of our future research efforts. For this reason, we feel it is not necessary to further expand on this limitation in the current discussion.

(2) The majority of the conclusions are derived primarily from the proteomic analyses. Although well conducted, it would strengthen the study to corroborate some of the major findings by other means such as IHC/IF with the corresponding quantifications and not only representative images. For example, the IF stainings for ECM1 should also be quantified - ECM1.

To strengthen our MS findings on ECM1 expression and to address the reviewer's concern, we have now included quantification of ECM1 using IF staining at selected time points in Figure S7E and we refer to these data in the Results section (p. 12 of the current manuscript). The IF quantification data correspond well to the MS data showing increase in ECM1 expression with fibrosis development and decline with partial fibrosis resolution.

(3) S1 - it would be important to show Sirius Red images over the time course, especially for CCl4 T4 where fibrosis resolution is occurring. Proteomics data also show this group clusters more closely with control mice and seeing a representative image would add further credibility to this point.

Requsted Sirius Red images are now part of the Figure S1B, documenting partial fibrosis resolution and overall parenchyma healing in T4 in both models.

(4) How comparable are the periods of the two models? 2 weeks in one model may not be the same as 2 weeks in the other depending on the severity of the pathogenesis.

We appreciate the reviewer’s comment regarding the comparability of time points between the two models. Indeed, the temporal dynamics of fibrosis development differ between the models employed in our study, and we have carefully considered this aspect to ensure the validity of our comparative analysis. To address this, we started our comparisons at a stage corresponding to the onset of fibrosis in each model. Specifically, quantification of Sirius Red-positive areas, indicative of collagen deposition (Figure S1B), revealed that 2 weeks of DDC treatment produced a comparable extent of fibrosis to that observed after 3 weeks of CCl₄ treatment. This point was designated as the initial fibrosis time point (T1, Figure S1B), from which further treatment was applied to induce more advanced fibrosis. This approach allowed us to standardize the comparison of fibrosis progression between the two models.

(5) Figure 4A-D - cell-type-specific signatures should be corroborated by actual IHC or IF stainings if possible. HNF4a (hepatocytes), CK19 (cholangiocytes), aSMA (activated fibrogenic HSCs), immune cells (B220, F4/80, Cd11b, CD11c etc).

We thank the reviewer for this valuable suggestion. To strengthen our analysis, we have now complemented the box plots of cell type-specific signatures derived from the MS data (Figure 4A-D) with immunofluorescence (IF) staining, which has been included in the Supplemental Data (Figure S6). Specifically, we provide representative IF images from control and T1-T4 time points for each model, documenting the changes in abundance with treatment in:

A) Hepatocytes (HNF4α), activated hepatic stellate cells (αSMA), and cholangiocytes (CK19).

B) Immune cell populations, including B cells (B220) and macrophages/monocytes/Kupffer cells (F4/80), as these immune cell groups were not only identified in our MS analysis but also have established roles in the selected models(3, 4, 5).

The representative images shown in Figure S6 show the dynamics of the cellular populations in each of the models, which correspond well with the MS data (compare Figures 4A-D and S5). These additional data further validate our findings and enhance the robustness of our conclusions.

References:

(1) Thiele M, Villesen IF, Niu L, et al. Opportunities and barriers in omics-based biomarker discovery for steatotic liver diseases. J Hepatol 2024;81:345-359.

(2) Friedman SL, Pinzani M. Hepatic fibrosis 2022: Unmet needs and a blueprint for the future. Hepatology 2022;75:473-488.

(3) Best J, Verhulst S, Syn WK, et al. Macrophage Depletion Attenuates Extracellular Matrix Deposition and Ductular Reaction in a Mouse Model of Chronic Cholangiopathies. PLoS One 2016;11:e0162286.

(4) Aoyama T, Inokuchi S, Brenner DA, et al. CX3CL1-CX3CR1 interaction prevents carbon tetrachlorideinduced liver inflammation and fibrosis in mice. Hepatology 2010;52:1390-400.

(5) Yang W, Chen L, Zhang J, et al. In-Depth Proteomic Analysis Reveals Phenotypic Diversity of Macrophages in Liver Fibrosis. J Proteome Res 2024;23:5166-5176.